# Learning Global Hypothesis Space for Enhancing Synergistic Reasoning Chain

**Jiaquan Zhang[1], Chaoning Zhang[1], Shuxu Chen[2], Xudong Wang[1], Zhenzhen Huang[1], Pengcheng Zheng[1]**
**Shuai Yuan[1], Sheng Zheng[1*], Qigan Sun[1], Jie Zou[1], Lik-Hang Lee [3], Yang Yang[1]**

[1]University of Electronic Science and Technology of China
[2]Kyung Hee University
[3]The Hong Kong Polytechnic University

## Abstract

Chain-of-Thought (CoT) has been shown to significantly improve the reasoning accuracy of large language models (LLMs) on complex tasks. However, due to the autoregressive, step-by-step generation paradigm, existing CoT methods suffer from two fundamental limitations. First, the reasoning process is highly susceptible to early-stage errors, which tend to propagate and amplify without a global coordination and correction mechanism, thereby distorting the overall reasoning chain. Second, current CoT methods lack structured analytical frameworks for pruning redundant reasoning and identifying critical reasoning features, resulting in instability and reduced interpretability. To address these issues, we propose Global Hypothesis Structure via Topological Data Analysis (GHS-TDA), which constructs a semantically enriched global hypothesis graph that integrates and coordinates multiple candidate reasoning paths, thereby supporting global consistency refinement and error mitigation. GHS-TDA applies persistent homology-based topological data analysis to capture stable multi-scale structures, remove redundancy and inconsistencies, and extract a more reliable reasoning skeleton. By jointly leveraging reasoning diversity and topological stability, GHS-TDA achieves self-adaptive convergence, produces high-confidence and interpretable reasoning paths, and consistently outperforms strong baselines in terms of both accuracy and robustness across multiple reasoning benchmarks.

## 1 Introduction

LLMs have demonstrated remarkable potential in tasks such as logical reasoning, mathematical proof, and multi-hop question answering. A representative technique to elicit such capabilities is CoT prompting (Wei et al., 2022), which improves interpretability and accuracy by decomposing complex problems into coherent intermediate steps.

Despite these advances, CoT and its extensions fundamentally rely on an autoregressive, step-by-step generation paradigm, where each reasoning step conditions on previously generated outputs Cao et al. (2026). Such sequential dependency increases the susceptibility of the reasoning process to early-stage errors, which may propagate and compound throughout the reasoning trajectory Wang et al. (2025). Moreover, the incremental generation mechanism lacks structured analytical control to regulate redundancy and identify salient reasoning features Chu et al. (2024), resulting in unstable reasoning trajectories and limited interpretability. To alleviate these limitations, a series of structured extensions have been proposed. Approaches such as Tree-of-Thought (ToT) (Yao et al., 2023a), Graph-of-Thought (GoT) (Besta et al., 2024), and Atom-of-Thought (AoT) (Teng et al., 2025) expand the reasoning space beyond single-path CoT and increase reasoning diversity. However, while these methods expand the reasoning space, their ability to mitigate autoregressive de-

---

*∗ Corresponding author. Contact: zszhx2021@gmail.com.
Additional contacts: jiaquanzhang2005@gmail.com, chaoningzhang1990@gmail.com,
ccccsx322@gmail.com, wl200203@khu.ac.kr, alley10086@gmail.com,
zpc777@std.uestc.edu.cn, sunqigan0206@gmail.com,
jie.zou@uestc.edu.cn, yang.yang@uestc.edu.cn, lik-hang.lee@polyu.edu.hk

pendency remains limited. Interactions across hypotheses are not explicitly coordinated, which may allow errors to accumulate across branches. Other frameworks attempt to enhance efficiency and robustness. ReAct (Yao et al., 2023b) integrates reasoning with acting, AFlow (Zhang et al., 2024) applies Monte Carlo tree search to reasoning workflows, and ReCEval (Prasad et al., 2023) evaluates reasoning chains for correctness and informativeness. While these methods improve task-level performance, they primarily emphasize outcome optimization rather than structural regulation, and thus lack a unified analytical framework for systematically characterizing reasoning chains. As explicit representations of the reasoning process Xia et al. (2025), reasoning chains inherently encode the semantic and logical organization of problem solving. However, without a principled structural perspective to characterize the topology and dependency patterns within reasoning chains, global coordination and consistency regulation remain underdeveloped. As a result, error propagation and redundant reasoning cannot be systematically controlled, limiting the reliability and interpretability of reasoning outcomes.

To address these challenges, this work introduces a new perspective: the reliability of reasoning depends not only on the correctness of locally generated results but also on the structural robustness exhibited by candidate paths within the global solution space. Unlike local heuristics, we adopt a topological perspective to model the reasoning space. Fundamentally, the reasoning space constitutes a high-dimensional complex structure formed by multiple interdependent candidate paths, which cannot be fully characterized by local indicators such as confidence scores or shortest-path length alone. TDA is capable of capturing stable connectivity and cyclic patterns across multiple scales, thereby providing a noise-insensitive and globally coherent structural measure. This perspective enables us to formalize concepts such as "logical backbones" and "self-consistent loops" as topological invariants, offering a principled basis for selecting and composing reasoning paths.

We propose GHS-TDA (Global Hypothesis Space with Topological Data Analysis), a two-stage framework that first constructs a global hypothesis graph through multi-role interactions to integrate diverse reasoning paths, and then applies topological analysis via persistent homology to extract stable backbones and self-consistent loops for interpretable reasoning chains. In the construction stage, we introduce a multi-role agenda mechanism consisting of explorers, verifiers, and bridges to dynamically generate and optimize a Global Hypothesis Graph (GHS). This process enables systematic integration and interaction of diverse reasoning information. Through unification, conflict detection, and closure inference, GHS enhances semantic connections among nodes and overcomes the isolation of traditional path-based generation. In the analysis stage, we leverage TDA (Munch, 2017; Chazal & Michel, 2021), specifically persistent homology, to extract robust reasoning skeletons and self-consistent cycles from the GHS. Analyzing the persistence of connected components ($H_0$) and loops ($H_1$) allows us to systematically capture backbone reasoning paths and self-verification structures, ultimately yielding reasoning chains with high confidence and interpretability.

Our key contributions are as follows:

- We introduce TDA into the reasoning chain, leveraging its scale invariance and structural robustness to provide a new perspective for analyzing and improving complex reasoning.

- We propose the GHS-TDA framework, a two-stage automated paradigm: the construction stage builds a Global Hypothesis Graph (GHS) via multi-role agenda mechanisms, and the analysis stage employs persistent homology with Betti stability checks to extract robust $H_0$ backbones and $H_1$ loops.

- We validate GHS-TDA on benchmarks including GSM8K, MATH, OlympiadBench, HotpotQA, MuSiQue, BBH, and LongBench, where it consistently outperforms existing methods in accuracy, consistency, and interpretability.

## 2 RELATED WORK

### 2.1 LLM REASONING OPTIMIZATION

AI and LLMs Zheng et al. (2025); Brown et al. (2020); Zheng et al. (2026b) demonstrate remarkable potential in complex reasoning tasks such as mathematical problem solving, logical deduction, and multi-hop question answering. However, their performance still heavily depends on carefully designed prompting strategies and reasoning structures (Achiam et al., 2023; Vaswani et al., 2017;

Zheng et al., 2026a). A seminal advance in this direction is Chain-of-Thought (CoT) prompting (Wei et al., 2022), which shows that decomposing complex problems into explicit intermediate steps significantly improves both the accuracy and interpretability of reasoning. This finding establishes prompting as a critical factor in eliciting reasoning capabilities from LLMs.

Building on CoT, researchers propose a range of structured extensions to further enrich the reasoning process. Tree-of-Thought (ToT) (Yao et al., 2023a), Graph-of-Thought (GoT) (Besta et al., 2024), and Atom-of-Thought (AoT) (Teng et al., 2025) introduce tree, graph, and atomic reasoning structures, respectively. These paradigms allow the model to explore multiple reasoning branches in parallel, reuse evidence across paths, and dynamically adjust reasoning trajectories, thereby alleviating the limitations of single-path CoT reasoning. Beyond structural extensions, frameworks such as ReAct (Yao et al., 2023b) and AFlow (Zhang et al., 2024) further integrate reasoning with external actions or search mechanisms. By combining reasoning with environment interaction or systematic search, these methods achieve stronger robustness and higher efficiency in complex tasks such as multi-hop QA and interactive problem solving.

Despite these advances, existing approaches still rely primarily on local heuristics for path selection and conflict resolution. They lack mechanisms for globally integrating diverse hypotheses or systematically analyzing the structural properties of reasoning chains, such as connectivity, consistency, and redundancy (Wang et al., 2022). This limitation often leads to fragmented reasoning, redundant exploration, or unstable convergence, especially in tasks that require reconciling multiple sources of evidence. Addressing these challenges motivates the development of new frameworks that combine global integration mechanisms with principled analytical tools for structural reasoning evaluation.

## 2.2 APPLICATIONS OF TOPOLOGICAL DATA ANALYSIS

TDA provides a powerful and principled framework for the structured analysis of high-dimensional data. Its core technique, persistent homology, captures the evolution of connected components and loops across multiple scales, thereby extracting structural features that remain stable under noise and local perturbations Munch (2017); Chazal & Michel (2021). Over the past decade, TDA has achieved successful applications in diverse fields, including bioinformatics Nicolau et al. (2011), material science Hiraoka et al. (2016), and neural network analysis Rieck et al. (2018); Naitzat et al. (2020). Beyond these domains, TDA also demonstrates broad potential in feature extraction, representation learning, and robustness evaluation within machine learning pipelines Hofer et al. (2017); Carrière et al. (2020).

However, its potential for reasoning research remains largely unexplored (Munch, 2017; Chazal & Michel, 2021). Reasoning chains produced by LLMs naturally exhibit graph-structured or sequential semantic and logical organization. Analyzing their topological properties through TDA offers the opportunity to identify stable connected components and self-consistent loops that persist across scales. These structures can then be mapped to backbone reasoning paths and consistency mechanisms, providing a principled way to filter redundant hypotheses, highlight critical connections, and improve both the stability and interpretability of reasoning processes. This perspective opens up a new line of inquiry into how topological robustness can complement semantic reasoning in LLMs.

## 3 METHOD

We propose **GHS-TDA**, a two-stage "construct–analyze" reasoning framework (Figure 1). In the *construction stage*, multiple reasoning paths sampled from an LLM are semantically aligned and merged into a unified Global Hypothesis Graph (GHS), which systematically integrates diverse information and manages conflicts. In the *analysis stage*, topological data analysis (TDA) is applied to extract stable backbones and self-consistent loops from the GHS, yielding high-confidence and interpretable reasoning paths. The construction ensures coherent integration, while the analysis exploits topological stability as a structural constraint and convergence criterion.

### 3.1 GLOBAL HYPOTHESIS SPACE MODELING

**Problem setup.** Given a problem $Q$, we first sample $N$ candidate reasoning paths

$$\mathcal{P} = \{P_1, \ldots, P_N\}, \quad P_i = (s_1^{(i)}, s_2^{(i)}, \ldots, s_{m_i}^{(i)}), \tag{1}$$

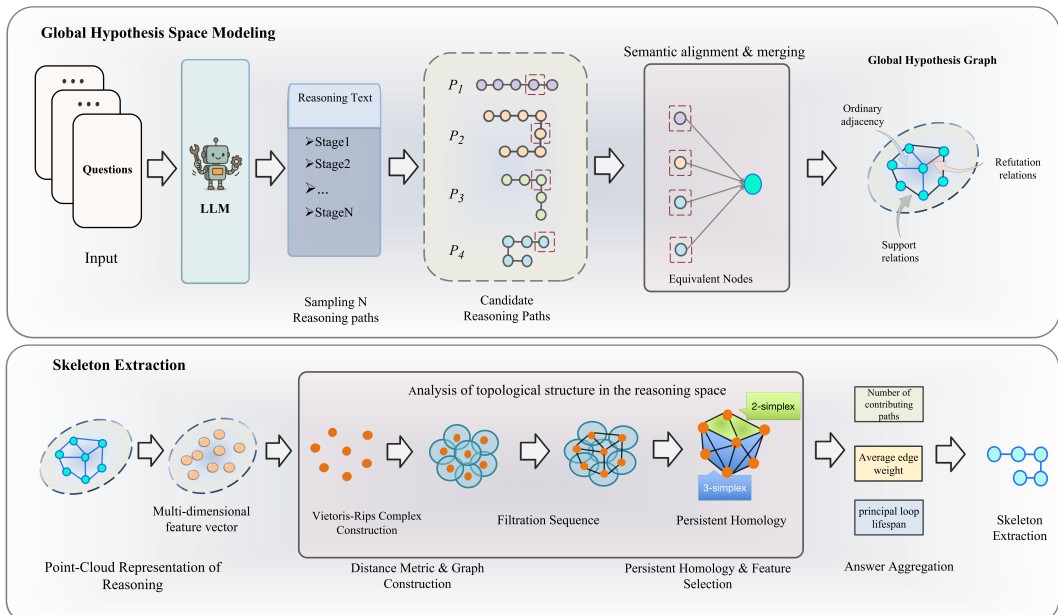

Figure 1: The method consists of two stages: (1) Global Hypothesis Space Modeling, where multiple reasoning paths sampled from an LLM are semantically aligned and merged into a unified Global Hypothesis Graph encoding adjacency, support, and refutation relations; and (2) Skeleton Extraction, where the graph is embedded into a feature space, analyzed via Vietoris–Rips filtration and persistent homology, and reduced to stable backbones and self-consistent loops. The resulting skeleton provides both accurate answers and interpretable reasoning structures.

where each $P_i$ denotes a stepwise sequence of intermediate hypotheses with variable length $m_i$. These paths may differ substantially in surface form, semantic fidelity, and logical coverage. Our goal is to integrate them into a single global structure that preserves diversity while eliminating redundancy.

**Graph definition.** We define the *Global Hypothesis Graph* (GHG) as

$$G = (V, E). \tag{2}$$

Each node $v \in V$ is represented as

$$v = (\texttt{text}, \texttt{canon}, c, r), \tag{3}$$

where: (i) $\texttt{text}$ stores the natural–language expression of the step; (ii) $\texttt{canon}$ is its canonicalized form (e.g., symbolic or normalized logical representation) used for equivalence testing; (iii) $c \in [0, 1]$ is the confidence score estimated from the LLM or aggregated statistics; (iv) $r \in [0, 1]$ is a normalized progress indicator reflecting how far the step is from the final answer. Edges $e = (v_i, v_j) \in E$ represent semantic–logical dependencies, typically arising from path adjacency, explicit usage (e.g., $s_i$ uses $s_j$), or inferred support/refutation. We elaborate the definition of "support" and "refutation" in the Appendix A.2.

This construction yields a directed multigraph in which all hypotheses generated by the model are placed into a shared reasoning space.

**Node alignment and merging.** A central step is to align semantically equivalent hypotheses across different paths. For two nodes $s_a$ and $s_b$, we compute the similarity of their canonicalized forms. If

$$\text{Sim}(\text{canon}(s_a), \text{canon}(s_b)) > \theta_{\text{merge}}, \tag{4}$$

the two nodes are merged into a single representative vertex, inheriting all incident edges. This merging criterion ensures that semantically equivalent reasoning steps, possibly expressed in different surface forms (e.g., "$2 + 2 = 4$" vs. "the sum is four"), are unified.

After merging, the confidence $c$ of the resulting node is computed as the average of its sources, while the progress $r$ is assigned as the maximum progress value among them to preserve downstream

completeness. We also maintain a record of provenance (i.e., which original paths contributed) to enable later attribution and evidence tracking.

**Resulting properties.** The resulting graph $G$ compactly encodes the union of all sampled reasoning paths without duplication, while retaining their semantic and logical structure. It preserves alternative hypotheses in a unified space, allowing systematic comparison of competing reasoning attempts and their interdependencies. At the same time, it provides a coherent foundation for subsequent topological analysis, where connected clusters naturally correspond to stable reasoning backbones and cycles capture self-consistent or cross-validating structures within the reasoning process.

## 3.2 SKELETON EXTRACTION

**Point-cloud representation.** Each node $v$ is embedded into a joint feature vector:

$$\mathbf{z}_v = \left[\, \mathbf{e}_v \,\|\, \boldsymbol{\phi}_{\text{graph}}(v) \,\|\, u_v \,\right], \tag{5}$$

where: (i) $\mathbf{e}_v$ is an L2-normalized semantic embedding; (ii) $\boldsymbol{\phi}_{\text{graph}}(v)$ encodes graph structure (progress $r_v$, BFS-based positional encoding, centrality), standardized per instance; (iii) $u_v = -\log(\text{confidence}_v + 10^{-6})$ denotes uncertainty. All features are normalized to ensure balanced contributions.

**Distance metric and filtration.** We define a mixed distance:

$$d(v_i, v_j) = \alpha\big(1 - \langle \mathbf{e}_i, \mathbf{e}_j \rangle\big) + \beta \,\|\boldsymbol{\phi}_{\text{graph}}(i) - \boldsymbol{\phi}_{\text{graph}}(j)\|_1 + \nu\,(u_i + u_j). \tag{6}$$

A $k$-nearest-neighbor graph ($k \approx 15$) is constructed and pruned by a global threshold $\tau$ (95th percentile of distances). A Vietoris–Rips filtration is then built on this sparsified graph, which preserves salient topological features ($H_0, H_1$) while reducing complexity.

**Persistent homology and feature selection.** We compute persistent homology up to $H_1$, obtaining barcodes for connected components ($H_0$) and loops ($H_1$). Significant features are selected by lifespan $L = \text{death} - \text{birth}$ (Top-$q\%$), with $H_0$ capturing major clusters and $H_1$ reflecting self-consistent loops.

**Operating scales and skeleton construction.** To map features back into the graph, we precisely define operating thresholds:

$$\varepsilon_{H_0} = \text{median}\{\text{death}(b) \mid b \in B_0\}, \qquad \varepsilon_b = 0.99 \cdot \text{death}(b),\ \forall b \in B_1. \tag{7}$$

This yields a thresholded subgraph $\mathcal{G}(\varepsilon)$ for cluster and loop analysis:

- *Clusters and anchors.* On $\mathcal{G}(\varepsilon_{H_0})$, we retain components $C$ with $|C| > 3$ that cover at least two reasoning paths. Anchors are chosen as

$$s_C = \arg\min_{v \in C} r_v, \quad g_C = \arg\max_{v \in C} r_v, \tag{8}$$

corresponding to start and goal nodes.

- *Loop assignment.* Each loop $b \in B_1$ is localized at $\varepsilon_b$ and assigned to the cluster with maximal overlap:

$$C(b) = \arg\max_{C} |V_b \cap C|. \tag{9}$$

- *Skeleton backbone.* For each $C$, we compute the shortest path $\mathcal{P}_C$ from $s_C$ to $g_C$ as the backbone. If a principal loop $b_C$ is assigned, we reroute via a pivot near median progress:

$$s_C \rightarrow v \rightarrow (\text{tour of } b_C) \rightarrow v \rightarrow g_C, \tag{10}$$

explicitly embedding a verification loop to enhance self-consistency. Loops are instantiated by a minimum-weight cycle basis (Horton's algorithm) or by stitching heuristics if fragmented.

When multiple clusters are available, we prioritize: (i) clusters with principal loops; (ii) larger loop lifespan; (iii) larger cluster size; (iv) smaller backbone cost.

**Answer aggregation.** We use confidence/persistence-weighted voting to get candidate answers aggregating along the skeleton. If loops are present, additional numeric substitution or entailment

checks are applied. The final output includes the high-confidence answer, skeleton structure, and key statistics (e.g., contributing paths, average edge weight, loop lifespan).

**Implementation details.** Embeddings: `text-embedding-3-large`; persistent homology: GUDHI (VR up to $H_1$); random seeds: 5; temperature: 0.7; top-p: 0.95; maximum LLM calls: 16 per example. Settings are fixed across baselines unless specified.

# 4 EXPERIMENT

## 4.1 EXPERIMENTAL SETUP

**Models.** We select three representative LLMs as backbones for reasoning: GPT-4o-mini, Qwen-Turbo, and DeepSeek-V3. These models differ in architecture and optimization strategies, which reduces bias from model-specific behaviors. All experiments run under unified decoding and budget constraints to ensure comparability.

**Baseline models.** We compare GHS-TDA with nine representative baselines that cover chain-based, tree-based, graph-based, forest-based, and atomic reasoning paradigms: CoT (Wei et al., 2022) and its self-consistency variant CoT-SC (Wang et al., 2022), Self-Refine (Madaan et al., 2023), Analogical Prompting (Yasunaga et al., 2023), the search-based framework AFlow (Zhang et al., 2024), and the structured approaches ToT (Yao et al., 2023a), GoT (Besta et al., 2024), FoT (Bi et al., 2024), and AoT (Teng et al., 2025). Together, these baselines provide a comprehensive benchmark for systematic comparison.

**Datasets.** We adopt eight widely used benchmarks covering arithmetic, mathematics, multi-hop, and long-context reasoning: GSM8K Cobbe et al. (2021), MATH Hendrycks et al. (2021a), Olympiad-Bench Zheng et al. (2024), BBH Srivastava et al. (2022), MMLU-CF Hendrycks et al. (2021b), LongBench Bai et al. (2023), HotpotQA Yang et al. (2018), and MuSiQue Trivedi et al. (2022)

**Evaluations.** We report Exact Match (EM, %) and four auxiliary metrics. For interpretability, three trained annotators rate clarity, logical coherence, credibility, and conciseness on a 1–5 Likert scale following a written rubric (IAA reported via Krippendorff's $\alpha$). Node confidence is the model-reported step probability calibrated on a held-out set; Confidence Stability is the standard deviation across steps on a path (lower is better). Computation Cost is the average number of LLM calls per problem. Statistical significance is assessed via paired $t$-tests against AoT with $\alpha = 0.05$ (per-dataset details in Appendix).

## 4.2 MAIN RESULTS

We evaluate GHS-TDA on eight reasoning and question-answering benchmarks, namely MATH, OlympiadBench, GSM8K, BBH, MMLU-CF, LongBench, HotpotQA, and MuSiQue. The comparison involves nine representative baselines: CoT, CoT-SC, Self-Refine, Analogical Prompting, AFlow, ToT, GoT, FoT, and AoT. Experiments are conducted across three backbone models: GPT-4o-mini, Qwen-Turbo, and DeepSeek-V3. The evaluation metric is exact match (EM) accuracy.

As shown in Table 1, GHS-TDA consistently delivers the best or near-best results across datasets and backbones. On `GPT-4o-mini`, it achieves 83.9% on `MATH`, surpassing AoT by 0.3 percentage points and CoT by 5.6 points. On `HotpotQA`, it reaches 81.4%, improving over AFlow by nearly eight points. On `MuSiQue`, it obtains 39.8%, outperforming ToT by 0.7 points and AoT by 1.4 points. On `Qwen-Turbo`, GHS-TDA achieves 87.9% on `BBH`, exceeding GoT by 3.0 points and AoT by 2.5 points, and reaches 80.3% on `HotpotQA`, a gain of over seven points compared to AFlow. On `DeepSeek-V3`, it records 14.7% on `OlympiadBench`, surpassing GoT by one point, and achieves 81.7% on `HotpotQA`, slightly higher than AoT at 80.6%. In terms of overall performance, GHS-TDA attains average EM scores of 68.0% on `GPT-4o-mini`, 67.6% on `Qwen-Turbo`, and 68.3% on `DeepSeek-V3`. These values consistently surpass the strongest baselines, with AoT reaching 66.9%, 66.8%, and 67.3% under the same settings. This demonstrates that the proposed global hypothesis–space framework outperforms representative multi-path reasoning methods in overall accuracy.

Table 1: Performance comparison across datasets (EM %).

| Method | MATH | OlympiadBench | gsm8k | BBH | MMLU-CF | LongBench | HotpotQA | MuSiQue | Avg |
|---|---|---|---|---|---|---|---|---|---|
| GPT-4o-mini | | | | | | | | | |
| CoT | 78.3 | 9.3 | 90.9 | 78.3 | 69.6 | 57.6 | 67.2 | 34.1 | 60.7 |
| CoT-SC ($n=5$) | 81.8 | 10.2 | 92.0 | 83.4 | 71.1 | 58.6 | 66.2 | 33.8 | 62.1 |
| Self-Refine | 78.7 | 9.4 | 91.7 | 80.0 | 69.7 | 58.2 | 68.3 | 35.1 | 61.4 |
| Analogical Prompting | 65.4 | 6.5 | 87.2 | 72.5 | 65.8 | 52.9 | 64.7 | 32.8 | 56.0 |
| AFlow | 83.0 | 12.4 | 93.5 | 76.0 | 69.5 | 61.0 | 73.5 | 38.1 | 63.4 |
| ToT | 79.2 | 11.4 | 94.9 | 84.1 | 69.9 | 62.8 | 76.8 | 39.1 | 64.8 |
| GoT | 83.0 | 13.1 | 94.5 | 85.9 | 70.2 | 63.1 | 74.2 | 36.5 | 65.1 |
| FoT ($n=8$) | 82.5 | 12.5 | 94.0 | 82.4 | 70.6 | 59.1 | 66.7 | 35.8 | 63.0 |
| AoT | 83.6 | 12.1 | 95.0 | 86.0 | 70.9 | 68.5 | 80.6 | 38.4 | 66.9 |
| **GHS-TDA (Ours)** | **83.9** | **14.5** | **95.2** | **88.4** | **71.6** | **69.5** | **81.4** | **39.8** | **68.0** |
| Qwen-Turbo | | | | | | | | | |
| CoT | 78.1 | 8.9 | 90.7 | 78.1 | 69.4 | 57.3 | 66.8 | 33.6 | 60.4 |
| CoT-SC ($n=5$) | 81.4 | 9.9 | 91.5 | 83.2 | 70.8 | 58.4 | 65.9 | 33.5 | 61.8 |
| Self-Refine | 78.5 | 9.4 | 91.4 | 79.8 | 69.5 | 58.0 | 68.2 | 35.0 | 61.2 |
| Analogical Prompting | 65.2 | 6.2 | 87.0 | 72.2 | 65.2 | 52.7 | 64.5 | 32.6 | 55.7 |
| AFlow | 82.4 | 12.1 | 93.1 | 75.7 | 69.3 | 60.4 | 73.2 | 37.8 | 63.0 |
| ToT | 78.9 | 11.3 | 94.2 | 83.7 | 69.6 | 62.4 | 76.4 | 38.4 | 64.4 |
| GoT | 82.7 | 13.0 | 93.8 | 84.9 | 70.1 | 62.8 | 74.0 | 36.4 | 64.7 |
| FoT ($n=8$) | 82.2 | 12.3 | 93.9 | 82.3 | 70.4 | 59.0 | 66.4 | 35.8 | 62.8 |
| AoT | 83.5 | 12.6 | 94.7 | 85.4 | 70.5 | 68.1 | 80.0 | 39.2 | 66.8 |
| **GHS-TDA (Ours)** | **83.7** | **14.4** | **94.8** | **87.9** | **71.2** | **68.6** | **80.3** | **39.6** | **67.6** |
| DeepSeek-V3 | | | | | | | | | |
| CoT | 78.5 | 9.5 | 91.3 | 78.5 | 69.9 | 57.7 | 67.4 | 34.2 | 60.9 |
| CoT-SC ($n=5$) | 82.0 | 10.4 | 92.1 | 83.6 | 71.5 | 58.9 | 66.6 | 34.0 | 62.4 |
| Self-Refine | 78.9 | 9.5 | 91.9 | 80.4 | 70.1 | 58.4 | 69.1 | 35.1 | 61.7 |
| Analogical Prompting | 65.6 | 6.7 | 87.6 | 72.8 | 66.1 | 53.4 | 64.9 | 33.1 | 56.3 |
| AFlow | 83.4 | 12.5 | 93.6 | 76.4 | 69.8 | 61.4 | 74.0 | 38.2 | 63.7 |
| ToT | 79.1 | 11.6 | 95.0 | 84.4 | 70.4 | 63.2 | 76.9 | 39.4 | 65.0 |
| GoT | 83.2 | 13.7 | 94.5 | 86.2 | 70.3 | 63.4 | 74.2 | 36.7 | 65.3 |
| FoT ($n=8$) | 82.7 | 12.6 | 94.2 | 82.6 | 70.5 | 59.3 | 66.8 | 36.2 | 63.1 |
| AoT | 84.0 | 13.1 | 95.1 | 86.1 | 70.8 | 68.7 | 80.6 | 39.6 | 67.3 |
| **GHS-TDA (Ours)** | **84.5** | **14.7** | **95.2** | **88.7** | **71.6** | **69.9** | **81.7** | **40.1** | **68.3** |

Table 2: Comparison of different path selection strategies within the Global Hypothesis Graph (GHS), combining quantitative evaluation and human-centered interpretability assessment.

| Path Type | Accuracy % | Avg. Length | Avg. Conf. | Conf. Std ↓ | Clarity | Coherence | Credibility | Conciseness |
|---|---|---|---|---|---|---|---|---|
| Shortest Path (GHS) | 75.2 | 5.8 | 0.81 | 0.12 | 3.6 | 2.9 | 3.4 | 4.3 |
| Max-Confidence Path (GHS) | 82.1 | 11.5 | 0.93 | 0.21 | 4.1 | 4.2 | 4.3 | 3.9 |
| Human-Selected Path (GHS) | 83.6 | 9.2 | 0.88 | 0.07 | 4.5 | 4.6 | 4.7 | 4.4 |
| **TDA Skeleton (Ours)** | **83.9** | **8.7** | **0.90** | **0.07** | **4.4** | **4.5** | **4.7** | **4.3** |

## 4.3 PATH SELECTION ANALYSIS

We examine whether topology-aware path selection outperforms local confidence–based selection. As shown in Table 2, we compare four strategies on the MATH dataset: shortest path, max-confidence path, human-selected path, and the TDA Skeleton from our GHS-TDA framework. Evaluation considered both quantitative indicators—accuracy, path length, confidence, and stability—and human judgments of clarity, coherence, credibility, and conciseness.

The shortest path was most concise with an average of 5.8 steps, but accuracy dropped to 75.2 percent and confidence was unstable with a variance of 0.12. The max-confidence path reached the highest confidence of 0.93, yet required 11.5 steps and showed high variance of 0.21. The human-selected path balanced these trade-offs, achieving 83.6 percent accuracy with 9.2 steps and a stable variance of 0.07. The TDA Skeleton slightly outperformed it, with 83.9 percent accuracy, 8.7 steps, and the same low variance, yielding compact and reliable chains.

Human evaluation aligned with these findings. The shortest path was concise but incoherent, the max-confidence path was moderately rated but verbose, while the human-selected path achieved the best overall scores. The TDA Skeleton closely matched human ratings, with clarity at 4.4, coherence at 4.5, credibility at 4.7, and conciseness at 4.3.

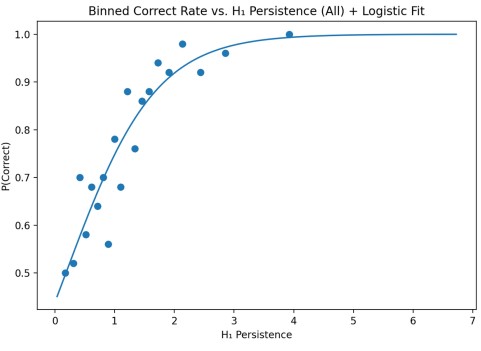 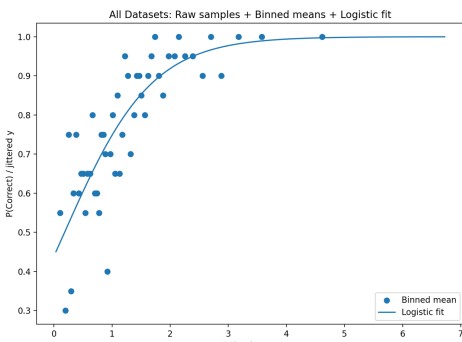

(a) Binned correct rate with logistic fit.

(b) Raw samples with binned means and logistic fit.

Figure 2: Global relation between $H_1$ persistence and reasoning correctness.

These results show that topological analysis enables automatic extraction of reasoning chains that are nearly as accurate and interpretable as those chosen by humans, while avoiding both under- and over-extension.

As shown in Table 3, we further evaluate the robustness of different path selection strategies under adversarial perturbations. Specifically, reasoning steps were paraphrased with semantically equivalent but lexically altered expressions to introduce local noise. Results show that the path selected by GHS-TDA achieves an

Table 3: Robustness under adversarial perturbations.

| Strategy | Before (%) | After (%) | Change (%) |
|---|---|---|---|
| Max-Confidence | 82.1 | 77.1 | 7.4 |
| **GHS-TDA (Ours)** | **83.9** | **81.5** | **2.9** |

accuracy drop of only 2.4 points with an answer change rate of 2.9%, significantly lower than the 7.4% observed for the Max-Confidence baseline. This indicates that paths identified by topological stability exhibit stronger internal logical connectivity and are less sensitive to superficial wording variations, whereas confidence-based paths are more vulnerable to semantic perturbations. These findings highlight the robustness advantage of structural evaluation beyond local heuristics.

## 4.4 ROBUSTNESS AND INTERPRETABILITY OF REASONING PROCESSES

We examine whether the proposed framework produces reasoning processes that are more robust and interpretable. We systematically evaluate the association between topological persistence and reasoning correctness across diverse tasks and difficulty levels. As shown in Fig. **??**, we analyze pooled samples from multiple datasets under a unified framework. The left panel aggregates instances into bins with a logistic regression fit, revealing a clear monotonic trend: higher persistence consistently predicts higher accuracy. The right panel confirms this result using raw samples with binned means, showing that the trend is robust and not an artifact of binning. This global analysis indicates that topological persistence serves as a principled, task-agnostic signal of reasoning reliability.

As shown in Fig. 3, we further validate the predictive value of topological persistence through both distributional and classification analyses. The boxplot analysis (Fig. 3a) demonstrates that correct reasoning chains consistently exhibit higher $H_1$ persistence values than incorrect ones, indicating that persistent topological structures capture stronger logical robustness. The ROC analysis (Fig. 3b) quantifies this effect, with persistence alone reaching an AUC of 0.74. These results confirm that $H_1$ persistence not only provides discriminative power but also enhances robustness and interpretability of reasoning processes. More detailed, per-dataset visualizations are presented in Appendix A.3, further illustrating the consistency of these findings across diverse benchmarks.

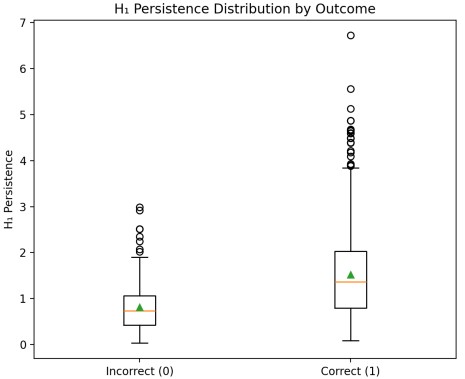

(a) Distribution of $H_1$ persistence by outcome.

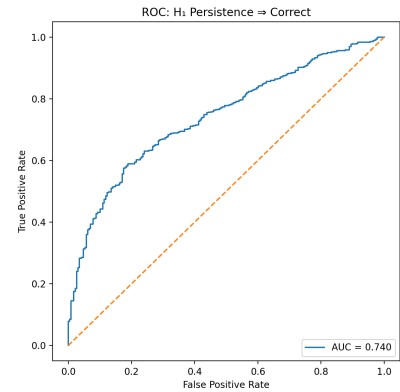

(b) ROC curve using $H_1$ persistence.

Figure 3: Validation of the predictive role of topological persistence. Left: correct reasoning chains have consistently higher $H_1$ persistence values than incorrect ones. Right: ROC analysis shows persistence alone achieves an AUC of 0.74.

Table 4: Predictive power of $H_1$ persistence for reasoning correctness. Higher persistence consistently correlates with better performance.

| Analysis Item | Value | Interpretation |
|---|---|---|
| Global Spearman $\rho$ | 0.349 ($p \approx 0$) | Moderate positive correlation |
| Logistic regression (std. $H_1$) | 1.247 (OR $\approx$ 3.48) | Strong effect: +1 SD $\Rightarrow \sim 3.5 \times$ odds |
| ROC–AUC ($H_1$ only) | 0.74 | Good discriminative ability |
| **Per-dataset ROC–AUC** | | |
| GSM8K | 0.748 | Robust |
| MATH | 0.704 | Robust |
| OlympiadBench | 0.703 | Robust |
| BBH | 0.729 | Robust |
| MMLU-CF | 0.733 | Robust |
| LongBench | 0.737 | Robust |
| HotpotQA | 0.778 | Strongest |
| MuSiQue | 0.709 | Robust |

## 4.5 Correlation Between Topology and Reasoning Accuracy

We investigate whether topological persistence is quantitatively associated with reasoning correctness. As shown in Table 4, $H_1$ persistence emerges as a strong predictor. A global Spearman correlation of 0.349 confirms a significant positive relationship: more persistent topological features correspond to higher accuracy. Logistic regression further shows that a one–standard deviation increase in persistence raises the odds of correctness by roughly 3.5 times, indicating a substantial effect size. Using persistence alone, ROC analysis yields an AUC of 0.74, demonstrating solid discriminative power.

This effect is consistent across all eight benchmarks. Per-dataset AUC values remain within the 0.70–0.78 range, with HotpotQA reaching 0.778. Such stability across arithmetic, symbolic, and multi-hop reasoning indicates that topological persistence provides a task-agnostic and statistically significant signal of reliability, offering a principled way to estimate correctness beyond ground-truth supervision.

## 4.6 Ablation Study

As shown in Table 5, we conduct an ablation study on the distance weights under the constraint $\alpha + \beta + \gamma = 1$. The full model with weights $(0.6, 0.3, 0.1)$ achieves an average accuracy of $83.9\%$. Removing the structural term ($\beta = 0$) reduces the accuracy to $81.2\%$, while removing the

semantic term ($\alpha = 0$) causes the largest drop, down to $77.4\%$, indicating its dominant contribution. Removing the uncertainty term ($\gamma = 0$) results in an accuracy of $83.5\%$, suggesting a smaller but consistent gain in robustness. Overall, semantic similarity is the most critical factor, structural features provide complementary benefits, and the uncertainty term, although lightweight, improves stability.

Table 5: Ablation study of distance weights under the constraint $\alpha + \beta + \gamma = 1$.

| Method | $\alpha$ | $\beta$ | $\gamma$ | Accuracy (Avg.) |
|---|---|---|---|---|
| **GHS-TDA (Ours)** | 0.6 | 0.3 | 0.1 | 83.9% |
| Without $\beta$ | 0.9 | 0.0 | 0.1 | 81.2% |
| Without $\alpha$ | 0.0 | 0.9 | 0.1 | 77.4% |
| Without $\gamma$ | 0.7 | 0.3 | 0.0 | 83.5% |

### 4.7 EFFICIENCY PERFORMANCE EXPERIMENT

As shown in Table 6, our method maintains a stable and fixed upper bound of 19 LLM calls across all tasks, substantially reducing inference cost compared with existing multi-path reasoning approaches. Under the same settings, GHS-TDA reduces the number of calls by approximately 25%–30% relative to ToT and 35%–40% relative to AoT, indicating that our discriminative relation inference effectively replaces the large amount of recursive generative evaluation used in prior work.

Since the actual computational cost of LLMs is more closely tied to token usage than to call count alone, we further report total token consumption in Table 7. The results show that GHS-TDA uses about 26.8% fewer tokens than ToT and 35.7% fewer tokens than AoT on average, confirming the efficiency advantage of our approach in reducing redundant generation and improving overall reasoning efficiency.

Table 6: Comparison of LLM Call Computational Costs

| Method | MATH | Olymp. | GSM8K | BBH | MMLU-CF | LongB. | HotpotQA | MuSiQue |
|---|---|---|---|---|---|---|---|---|
| CoT | 1 | 1 | 1 | 1 | 1 | 1 | 1 | 1 |
| CoT-SC ($n = 16$) | 16 | 16 | 16 | 16 | 16 | 16 | 16 | 16 |
| ToT | 25.8 | 27.2 | 13.5 | 24.1 | 19.3 | 14.2 | 20.7 | 26.4 |
| GoT | 21.6 | 23.5 | 14.8 | 22.9 | 18.4 | 13.9 | 17.8 | 20.3 |
| AoT | 29.7 | 32.4 | 17.2 | 28.6 | 24.9 | 16.8 | 23.5 | 29.1 |
| **GHS-TDA (Ours)** | **19** | **19** | **19** | **19** | **19** | **19** | **19** | **19** |

Table 7: Comparison of token consumption.

| Method | MATH | OlympiadBench | GSM8K | BBH | MMLU-CF | LongBench | HotpotQA | MuSiQue |
|---|---|---|---|---|---|---|---|---|
| CoT | 2290 | 4590 | 278 | 642 | 1235 | 83 | 1305 | 485 |
| CoT-SC | 36640 | 73440 | 4448 | 10272 | 19760 | 1328 | 20880 | 7760 |
| ToT | 88623 | 187272 | 5630 | 23208 | 35753 | 1768 | 40520 | 19206 |
| GoT | 64303 | 140225 | 5349 | 19112 | 29541 | 1500 | 30198 | 12799 |
| AoT | 68013 | 148716 | 4782 | 18361 | 30752 | 1394 | 30668 | 14114 |
| GHS-TDA | 51411 | 83309 | 7709 | 15793 | 28862 | 1685 | 28358 | 10263 |

## 5 CONCLUSION

In this work, we present GHS-TDA, a two-stage framework that integrates Global Hypothesis Graph construction with topological data analysis for robust reasoning. By unifying diverse reasoning paths into a coherent hypothesis space and extracting stable backbones and self-consistent loops via persistent homology, the framework improves both accuracy and interpretability. Experiments across multiple benchmarks demonstrate consistent gains over strong baselines, while analysis of topological persistence establishes it as a task-agnostic indicator of reasoning reliability. This work highlights the value of combining structural integration with topological robustness, providing a principled foundation for more reliable and transparent reasoning systems.

## 6 ACKNOWLEDGMENTS

This work was partially supported by the National Natural Science Foundation of China under grants 62220106008, and 62572104.

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

# A  APPENDIX

## A.1  ACKNOWLEDGE

This article used large language models (such as ChatGPT) as an auxiliary tool in the language polishing process, but did not use them in research conception and academic content generation.

## A.2  DEFINITION OF SUPPORT AND REFUTATION

To efficiently incorporate logical relations into the TDA distance, we avoid evaluating all $\mathcal{O}(|V|^2)$ node pairs. Instead, we construct a reduced candidate set $\mathcal{L} = \mathcal{L}_{\text{long}} \cup \mathcal{L}_{\text{lat}}$ consisting of two types of pairs. The longitudinal set $\mathcal{L}_{\text{long}}$ contains existing derivation edges $(v_i, v_j)$ in the GHG, which correspond to potential "premise $\rightarrow$ conclusion" relations. The lateral set $\mathcal{L}_{\text{lat}}$ includes unconnected node pairs $(v_a, v_b)$ that satisfy

$$|r(v_a) - r(v_b)| < \epsilon,$$

which typically represent competing hypotheses at the same reasoning stage and may potentially contradict each other.

The resulting candidate set has size $K \ll |V|^2$. To infer logical relations efficiently, we partition $\mathcal{L}$ into $C$ chunks of size $S$ (e.g., $S = 20$) and process each chunk with a single LLM call. For each node pair $(v_i, v_j)$, the model assigns one of three labels: SUPPORT, REFUTE, or NEUTRAL, yielding a logical code $R(v_i, v_j)$. The inferred logical relation is then integrated as an additional term into the TDA distance function:

$$d(v_i, v_j) = \alpha \left(1 - \langle e_i, e_j \rangle\right) + \beta \left\| \phi_{\text{graph}}(i) - \phi_{\text{graph}}(j) \right\|_1 + \nu(u_i + u_j) + \delta \cdot R(v_i, v_j).$$

Specifically, if a pair is labeled REFUTE, we set $R(v_i, v_j) = +M$, where $M$ is a large positive constant, thereby pushing the corresponding nodes far apart in the embedding space. If a pair is labeled SUPPORT, we set $R(v_i, v_j) = -W$ (e.g., $W = 1$), which draws the nodes closer and encourages logically coherent connections. If a pair is labeled NEUTRAL or does not belong to the candidate set, we set $R(v_i, v_j) = 0$, reducing the distance to its original form.

## A.3  PERSISTENCE–ACCURACY ANALYSIS ACROSS DATASETS

Figure 4 presents logistic fits of reasoning accuracy as a function of $H_1$ persistence, with binned means overlaid. Across all eight datasets, we consistently observe a monotonic increase, demonstrating that more persistent topological loops reliably predict higher correctness. The effect is most

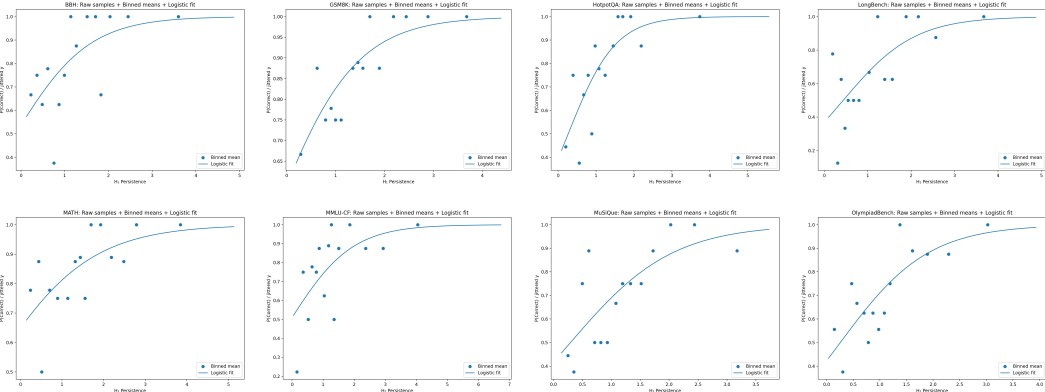

Figure 4: Overall results across eight experimental settings.

pronounced in the low-to-moderate persistence regime ($H_1 \in [0,2]$), where small increases in persistence correspond to sharp gains in accuracy. At higher values, curves gradually saturate, reflecting a ceiling effect once loop stability is sufficient.

**Arithmetic and short-chain reasoning.** On **GSM8K**, the curve rises steeply and quickly saturates near perfect accuracy. This suggests that persistent loops in arithmetic tasks effectively function as self-verification mechanisms (e.g., numeric substitution or equation consistency checks). Similarly, **MATH** exhibits a monotonic increase but with a smoother slope, indicating that more complex derivations require higher persistence levels to capture the complete reasoning backbone.

**Long-context and multi-hop reasoning.** Datasets such as **HotpotQA**, **LongBench**, and **MuSiQue** show the steepest slopes at low persistence and saturate earlier than other tasks. This pattern highlights the importance of stable loop structures for integrating multiple evidence sources and maintaining coherence across extended reasoning chains. **HotpotQA**, in particular, reaches near-perfect accuracy once persistence exceeds moderate values, reflecting the decisive role of structural self-consistency in cross-document reasoning.

**Knowledge-intensive tasks.** For **MMLU-CF**, persistence provides a strong positive signal, with accuracy steadily rising as loops become more stable. The trend indicates that persistence mitigates the effects of noisy or uncertain knowledge retrieval by reinforcing structurally coherent reasoning paths.

**Challenging and creative reasoning.** **OlympiadBench** exhibits a clear upward trend but with a slightly lower plateau compared to other datasets. This suggests that while loop persistence improves correctness, certain Olympiad-level problems involve creative steps or lengthy derivations that may not be fully captured by first-order topological features alone. Nonetheless, persistence remains a robust predictor of accuracy.

**Summary.** Taken together, these results confirm that $H_1$ persistence serves as a task-agnostic reliability signal across diverse benchmarks. In arithmetic tasks it captures self-verification, in long-context reasoning it enforces multi-evidence integration, and in knowledge-intensive settings it suppresses noisy paths. We therefore recommend persistence-aware path selection strategies, using thresholding (e.g., $H_1 \geq 1$) or weighted scoring, to enhance both robustness and interpretability of reasoning chains.

## A.4 STABILITY ANALYSIS

To assess the stability of the proposed GHS-TDA pipeline, we evaluate both its computational cost and the robustness of the resulting topological features. In our experiments on GSM8K and MATH, each problem instance produces roughly 80–120 nodes, with the KNN neighborhood size fixed at 15. End-to-end graph construction—including canonicalization, node merging, node embedding, and

KNN graph building—requires only 25–60 ms per instance on a system equipped with an RTX 4090 GPU and 32 GB of CPU memory, with embedding computation and KNN construction dominating the runtime. The subsequent Vietoris–Rips filtration, computed up to the first homology group, operates on these sparse graphs and exhibits a peak memory usage of 150–400 MB for graphs with around 100 nodes, and remains below 1 GB even when the graph size grows to approximately 200 nodes. Persistent-homology computation is also efficient: computing the zeroth and first homology groups with GUDHI takes 10–25 ms per instance, corresponding to only 10–30% of the overall pipeline time. These results indicate that the method scales well within typical LLM-reasoning workloads.

We further evaluate numerical stability by perturbing node embeddings with isotropic Gaussian noise of magnitude between 0 and 0.1 and recomputing the persistence diagrams. The resulting bottleneck distances lie in the range 0.01–0.10, depending on the noise level, and remain significantly smaller than the lifetimes of the dominant first-homology features, which typically range from 0.3 to 0.8. This demonstrates that the extracted topological signatures are highly robust to embedding perturbations, consistent with the theoretical stability properties of persistent homology. Overall, the pipeline exhibits low computational overhead, favorable scaling behavior, and strong robustness, supporting its suitability for large-scale multipath reasoning analysis.

## A.5 GENERALIZATION

The Table 8 shows that GHS-TDA delivers consistent and significant improvements on both models. On Llama 3-8B, it achieves an average score of 63.88, outperforming CoT, ToT, and AoT by 6.9, 3.0, and 1.1 points, respectively. On Qwen2-14B, it reaches 62.35, with gains of 6.75, 2.7, and 1.5 points over the corresponding baselines. Notably, these improvements are obtained without any additional supervision or external knowledge, indicating that the benefits come from the reasoning mechanism itself rather than model size. Overall, while all methods improve as the base model scales from 8B to 14B, the relative gains of GHS-TDA remain stable, demonstrating strong transferability and robustness across model sizes and architectures.

Table 8: Generalization ability of GHS-TDA.

| Backbone | Method | MATH | OlympiadBench | GSM8K | BBH | MMLU-CF | LongBench | HotpotQA | MuSiQue | Avg |
|---|---|---|---|---|---|---|---|---|---|---|
| | CoT | 72.04 | 8.37 | 87.26 | 74.38 | 65.42 | 53.57 | 63.17 | 31.71 | 56.99 |
| | ToT | 72.86 | 10.26 | 91.10 | 79.89 | 65.71 | 58.40 | 72.19 | 36.36 | 60.85 |
| Llama 3-8B | GoT | 76.36 | 11.79 | 90.72 | 81.61 | 65.99 | 58.68 | 69.75 | 33.95 | 61.10 |
| | AoT | 76.91 | 10.89 | 91.20 | 81.70 | 66.65 | 63.71 | 75.76 | 35.71 | 62.82 |
| | **GHS-TDA** | **77.19** | **13.05** | **91.39** | **83.98** | **67.30** | **64.64** | **76.52** | **37.01** | **63.88** |
| | CoT | 60.80 | 6.40 | 86.70 | 75.80 | 65.10 | 52.70 | 64.90 | 32.40 | 55.60 |
| | ToT | 65.00 | 7.20 | 89.80 | 79.70 | 67.60 | 59.30 | 70.80 | 37.90 | 59.66 |
| Qwen 2-14B | GoT | 65.70 | 8.40 | 89.10 | 80.40 | 67.80 | 58.10 | 72.10 | 35.40 | 59.62 |
| | AoT | 66.20 | 7.80 | 90.80 | 80.90 | 68.90 | 61.20 | 73.40 | 37.30 | 60.81 |
| | **GHS-TDA** | **68.40** | **9.70** | **91.20** | **82.50** | **69.80** | **62.80** | **75.80** | **38.60** | **62.35** |

We conducted additional experiments with GHG-TDA on different base large language models, using GPT-4o and Claude 3.5 Sonnet.

The Table 9 shows that GHG-TDA consistently achieves the best overall performance on both models, with an average score of 72.4 on GPT-4o and 72.8 on Claude 3.5 Sonnet, outperforming AoT, GoT, ToT, and CoT on all benchmarks. The consistent gains across different datasets and model backbones indicate that the effectiveness of GHG-TDA is not tied to a single powerful LLM, but instead stems from its graph-guided hierarchical reasoning mechanism. These results demonstrate the strong cross-model generalization capability of the proposed method.

## A.6 CORRELATION ANALYSIS

Tables 10 and 11 present the statistical analysis of the relationship between $H_1$ persistence and reasoning correctness. Table 10 shows correlation metrics across our primary dataset collection, while Table 11 provides the same analysis specifically for the deepseek-V3 model. The tables report global correlation measures and dataset-specific discrimination ability, quantifying how topological features in reasoning traces relate to successful problem-solving across diverse reasoning tasks.

Table 9: Performance comparison of different reasoning methods on two base models.

| Base Model | Method | MATH | OlympiadBench | GSM8K | BBH | MMLU-CF | LongBench | HotpotQA | MuSiQue | Avg |
|---|---|---|---|---|---|---|---|---|---|---|
| GPT-4o | CoT | 85.4 | 12.5 | 92.7 | 82.0 | 72.1 | 62.0 | 71.9 | 36.2 | 64.4 |
| | ToT | 86.9 | 14.1 | 96.0 | 89.1 | 73.0 | 66.0 | 82.1 | 40.5 | 68.5 |
| | GoT | 88.4 | 15.3 | 95.2 | 90.5 | 73.2 | 66.8 | 80.4 | 39.6 | 68.7 |
| | AoT | 89.1 | 15.0 | 96.5 | 92.0 | 74.3 | 72.6 | 86.0 | 42.1 | 71.0 |
| | GHG-TDA | 90.0 | 18.3 | 96.9 | 94.2 | 75.0 | 73.8 | 87.4 | 43.6 | 72.4 |
| Claude 3.5 Sonnet | CoT | 84.1 | 12.9 | 93.5 | 84.2 | 73.0 | 61.7 | 72.8 | 37.1 | 64.9 |
| | ToT | 86.0 | 14.8 | 96.2 | 90.8 | 74.1 | 66.4 | 83.3 | 41.5 | 68.9 |
| | GoT | 87.3 | 16.0 | 95.4 | 92.1 | 74.4 | 67.0 | 81.0 | 40.6 | 69.2 |
| | AoT | 88.0 | 15.7 | 96.8 | 93.4 | 75.2 | 72.2 | 86.7 | 43.2 | 71.3 |
| | GHG-TDA | 89.1 | 19.0 | 97.1 | 95.4 | 75.9 | 73.5 | 88.0 | 44.6 | 72.8 |

Table 10: Correlation between $H_1$ persistence and reasoning correctness across datasets.

| Analysis Item | Result | Interpretation |
|---|---|---|
| Global Spearman correlation ($\rho$) | 0.314 ($p \approx 0$) | Moderate correlation with correctness. |
| Logistic regression coefficient | 0.736 (OR $\approx$ 2.09 per ↑1SD) | +1SD nearly doubles correctness odds. |
| ROC-AUC ($H_1$ persistence only) | 0.671 | Good discrimination ability. |
| Dataset-level AUC | | Stable across tasks. |
| GSM8K | 0.699 | Robust. |
| MATH | 0.597 | Robust. |
| OlympiadBench | 0.686 | Robust. |
| BBH | 0.663 | Robust. |
| MMLU-CF | 0.703 | Robust. |
| LongBench | 0.764 | Robust. |
| HotpotQA | 0.617 | Robust. |
| MuSiQue | 0.627 | Robust. |

## A.7 PARAMETER SETTINGS AND TUNING STRATEGIES

Hyperparameter settings directly affect both the degree of structural compression in the Global Hypothesis Graph (GHG) and the sensitivity of the topological analysis. The key parameters include:

- node merging threshold $\theta_{\mathrm{merge}}$, - hybrid distance weights $(\alpha, \beta, \gamma)$, - number of significant topological features $K$, - loop embedding threshold $\delta$.

These are designed under the principle of *"semantics-dominant, structure-assisted, uncertainty-corrected"* reasoning.

**Hybrid distance construction.** We constrain

$$\alpha + \beta + \gamma = 1,$$

so that semantic similarity, structural consistency, and uncertainty are comparable on the same scale. The default setting is $(\alpha, \beta, \gamma) = (0.6, 0.3, 0.1)$. - Semantic similarity ($\alpha$) dominates clustering, thus receives the highest weight. - Structural consistency ($\beta$) is crucial in multi-hop or cross-document reasoning. - Uncertainty ($\gamma$) down-weights low-confidence nodes and acts as regularization.

Tuning guideline: - Increase $\alpha$ for arithmetic or short logical inference. - Increase $\beta$ for long-chain or cross-document reasoning. - Increase $\gamma$ under noisy outputs or fluctuating confidence.

**Node merging threshold.** The default $\theta_{\mathrm{merge}} = 0.85$ balances redundancy removal and connectivity. - Too low ($< 0.8$): risk of merging non-equivalent expressions. - Too high ($> 0.9$): graph becomes sparse, losing connectivity. Empirically: - Use $0.75$–$0.8$ in noisy tasks, - Use $0.9$ in precise reasoning (math, code).

**Number of significant topological features.** We use $K = 5$ by default to capture diverse backbones without redundancy. - Too small $K$: omits plausible reasoning chains. - Too large $K$: introduces noisy cycles, weakening interpretability. Practical range: $K \in [3, 8]$, tuned by task complexity and candidate path size.

Table 11: Correlation analysis between $H_1$ persistence and reasoning correctness across datasets, deepseek-V3.

| Analysis Item | Result | Interpretation |
|---|:---:|---|
| Global Spearman correlation ($\rho$) | 0.391 ($p \approx 0$) | Moderate correlation with correctness. |
| Logistic regression coefficient | 1.046 (OR $\approx$ 2.847 per $\uparrow$1SD) | +1SD triples correctness odds. |
| ROC-AUC ($H_1$ persistence only) | 0.726 | Good discrimination ability. |
| Dataset-level AUC | | Consistent across tasks. |
| GSM8K | 0.641 | Robust discrimination. |
| MATH | 0.770 | Robust discrimination. |
| OlympiadBench | 0.689 | Robust discrimination. |
| BBH | 0.690 | Robust discrimination. |
| MMLU-CF | 0.791 | Robust discrimination. |
| LongBench | 0.770 | Robust discrimination. |
| HotpotQA | 0.741 | Robust discrimination. |
| MuSiQue | 0.734 | Robust discrimination. |

**Loop embedding threshold.** The default $\delta = 0.15$ controls which loops are embedded into the backbone. We further adopt adaptive scaling:

$$\delta = \lambda \cdot \epsilon_b, \quad \lambda \in [0.1, 0.2],$$

where $\epsilon_b$ is the persistence scale of loop features. - Large $\delta$: may introduce noisy, distant loops. - Small $\delta$: may discard important verification loops.

**Summary.** The default hyperparameters provide a balanced configuration for general tasks. Practical tuning follows the order: 1. Fix $\alpha + \beta + \gamma = 1$, redistribute according to task. 2. Tune $\theta_{\text{merge}}$ and $K$ via grid search. 3. Set $\delta$ adaptively using $\epsilon_b$ scaling.

This strategy ensures robustness and reproducibility across diverse reasoning scenarios.

## A.8 PARAMETER SETTINGS AND TUNING STRATEGIES

Hyperparameters affect both graph compression and topological sensitivity. The key ones are the node merging threshold $\theta_{\text{merge}}$, distance weights $(\alpha, \beta, \gamma)$, number of topological features $K$, and loop threshold $\delta$, designed under the principle of *"semantics-dominant, structure-assisted, uncertainty-corrected"* reasoning.

**Hybrid distance.** We constrain $\alpha + \beta + \gamma = 1$ with default $(0.6, 0.3, 0.1)$. Semantic similarity ($\alpha$) dominates, structure ($\beta$) supports long or multi-hop tasks, and uncertainty ($\gamma$) regularizes noise. Adjust by increasing $\alpha$ for precise reasoning, $\beta$ for long dependencies, and $\gamma$ for noisy outputs.

We tested various combinations of Hybrid distance. As shown in Table 12 our node representation integrates semantic embeddings, graph-structural features, and an uncertainty term. This design enables TDA to simultaneously capture semantic coherence, structural consistency, and potential sources of noise. The confidence component is defined as $u_v = -\log(\text{confidence})$ and is assigned a small weight in the hybrid distance metric ($\gamma = 0.1$). Thus, even if the confidence estimates are imperfect, their overall influence remains limited. Moreover, persistent homology is inherently robust to local perturbations: erroneous confidence values lead only to short-lived topological features, which are automatically filtered out and do not appear in the final skeleton. As a result, the final reasoning outcome is determined primarily by topological persistence and structural consistency rather than by confidence itself. Consequently, imperfect confidence scores neither distort node representations nor degrade.

**Node merging.** Default $\theta_{\text{merge}} = 0.85$ balances redundancy and connectivity. Use 0.75–0.8 for noisy tasks, 0.9 for precise domains (e.g., math, code).

Table 12: Study of the distance metric.

| $\alpha$ (Semantic) | $\beta$ (Structural) | $\gamma$ (Uncertainty) | Accuracy (%) |
|---|---|---|---|
| 0.60 | 0.30 | 0.10 | **83.9** |
| 0.90 | 0.00 | 0.10 | 77.4 |
| 0.00 | 0.90 | 0.10 | 75.2 |
| 0.70 | 0.30 | 0.00 | 82.6 |
| 0.53 | 0.27 | 0.20 | 83.4 |
| 0.70 | 0.20 | 0.10 | 83.7 |
| 0.50 | 0.40 | 0.10 | 82.9 |

We examine how the node-merging threshold influences the construction of the Global Hypothesis Graph. The results are shown in Table 13 A low threshold (around the 0.70 range) induces overly aggressive consolidation, where semantically different reasoning steps are forced into a single node. This leads to semantic collapse, reduced path diversity, and weakened topological structure, ultimately degrading accuracy. As the threshold moves into a moderate region (approximately 0.80–0.90), semantically aligned steps are merged reliably while the graph remains well connected. In this regime, the resulting structure preserves both diversity and coherence, enabling persistent H0 backbones and H1 verification loops to be detected consistently. This balance produces the most stable and accurate reasoning performance across datasets. When the threshold becomes overly conservative (around the 0.95 range), the graph becomes fragmented due to insufficient merging of near-equivalent steps. Such sparsity reduces the emergence of meaningful topological features and limits the ability of TDA to extract coherent reasoning skeletons. These observations collectively indicate that moderate thresholds naturally yield the best trade-off between semantic integration and structural robustness.

Table 13: Effect of the node-merging threshold $\theta_{\text{merge}}$ on EM accuracy.

| $\theta_{\text{merge}}$ | EM Accuracy (%) |
|---|---|
| 0.70 | 72.6 |
| 0.80 | 83.1 |
| 0.85 (default) | **83.9** |
| 0.90 | 83.4 |
| 0.95 | 78.3 |

**Topological features.** Default $K = 5$ captures diverse backbones without noise; practical range 3–8, tuned by task complexity.

**Loop threshold.** Default $\delta = 0.15$, with adaptive scaling

$$\delta = \lambda \cdot \epsilon_b, \ \lambda \in [0.1, 0.2],$$

where $\epsilon_b$ is loop persistence. Larger $\delta$ risks noisy loops; smaller may drop useful ones.

**Summary.** Defaults are balanced for general use. Tuning priority: (1) redistribute $(\alpha, \beta, \gamma)$; (2) grid search $\theta_{\text{merge}}, K$; (3) adapt $\delta$ via persistence scaling.

## A.9 PSEUDOCODE

---

**Algorithm 1** GHS-TDA: Construct–Analyze Pipeline

---

**Require:** Problem $Q$; number of sampled paths $N$; merge threshold $\theta_{\text{merge}}$; distance weights $(\alpha, \beta, \nu)$; KNN size $k$; truncation percentile $\tau$; topological feature budget $K$; loop-embedding threshold $\delta$

**Ensure:** Final answer $\hat{a}$; reasoning skeleton $\mathcal{S}$

1: $\mathcal{P} \leftarrow \text{SAMPLEPATHS}(Q, N)$       ▷ LLM-based sampling of candidate reasoning paths
2: $G \leftarrow (V, E) \leftarrow \text{BUILDGHG}(\mathcal{P}, \theta_{\text{merge}})$       ▷ Global Hypothesis Graph with merged equivalent nodes
3: $\{\mathbf{z}_v\}_{v \in V} \leftarrow \text{EMBEDNODES}(G)$       ▷ $\mathbf{z}_v = [\, \mathbf{e}_v \,\|\, \phi_{\text{graph}}(v) \,\|\, u_v \,]$
4: $d(\cdot, \cdot) \leftarrow \alpha(1 - \langle \cdot, \cdot \rangle) + \beta \| \cdot \|_1 + \nu(\cdot + \cdot)$       ▷ Hybrid distance over $(\mathbf{e}, \phi_{\text{graph}}, u)$
5: $\mathcal{G}_{\text{KNN}} \leftarrow \text{BUILDKNN}(\{\mathbf{z}_v\}, d, k)$; $\mathcal{G}_\tau \leftarrow \text{TRUNCATE}(\mathcal{G}_{\text{KNN}}, \tau)$
6: $\text{VR} \leftarrow \text{VIETORISRIPS}(\mathcal{G}_\tau, d)$       ▷ Filtration over sparsified metric graph
7: $(\mathcal{D}_{H_0}, \mathcal{D}_{H_1}) \leftarrow \text{PERSISTENTHOMOLOGY}(\text{VR})$       ▷ Persistence diagrams/barcodes
8: $B_0^*, B_1^* \leftarrow \text{SELECTTOPBYLIFESPAN}(\mathcal{D}_{H_0}, \mathcal{D}_{H_1}, K)$       ▷ Top-$K$ significant features
9: $\varepsilon_{H_0}^* \leftarrow \text{median}\{\text{death}(b) : b \in B_0^*\}$; $\forall b \in B_1^*: \varepsilon_b^* \leftarrow 0.99 \cdot \text{death}(b)$
10: $\mathcal{S} \leftarrow \text{EXTRACTSKELETON}(G, d, \varepsilon_{H_0}^*, \{\varepsilon_b^*\}, B_1^*, \delta)$
11: $\hat{a} \leftarrow \text{AGGREGATEANSWERS}(\mathcal{S})$       ▷ Confidence/persistence-weighted voting with validation
12: **return** $(\hat{a}, \mathcal{S})$

---

**Algorithm 2** BUILDGHG: Construct Global Hypothesis Graph with Node Alignment

---

**Require:** Paths $\mathcal{P} = \{P_i\}_{i=1}^N$; merge threshold $\theta_{\text{merge}}$

**Ensure:** Graph $G = (V, E)$

1: $V \leftarrow \emptyset, E \leftarrow \emptyset$
2: **for** $i = 1$ to $N$ **do**
3:      **for** $j = 1$ to $m_i$ **do**
4:          $s \leftarrow s_j^{(i)}$; $(\text{text}(s), \text{canon}(s), c(s), r(s)) \leftarrow \text{ANNOTATE}(s)$
5:          $v^\star \leftarrow \arg\max_{v \in V} \text{Sim}(\text{canon}(s), \text{canon}(v))$       ▷ Search best canonical match
6:          **if** $V = \emptyset$ **or** $\text{Sim}(\text{canon}(s), \text{canon}(v^\star)) \leq \theta_{\text{merge}}$ **then**
7:              $v_{\text{new}} \leftarrow (\text{text}(s), \text{canon}(s), c(s), r(s))$; $V \leftarrow V \cup \{v_{\text{new}}\}$
8:              $\text{INITPROVENANCE}(v_{\text{new}}, i, j)$
9:          **else**
10:              $v^\star \leftarrow \text{MERGE}(v^\star, s)$       ▷ Avg confidence; max progress; provenance union
11:          **end if**
12:          **if** $j > 1$ **then**
13:              Add $e = (v_{j-1}^{(i)}, v_j^{(i)})$ to $E$       ▷ Temporal/deductive edge along $P_i$
14:          **end if**
15:      **end for**
16: **end for**
17: **return** $(V, E)$

---

---

**Algorithm 3** EXTRACTSKELETON: Backbone and Loop Embedding

---

**Require:** Graph $G = (V, E)$; distance $d$; cluster scale $\varepsilon_{H_0}^*$; loop scales $\{\varepsilon_b^*\}$; loop set $B_1^*$; embedding threshold $\delta$
**Ensure:** Reasoning skeleton $\mathcal{S}$
1: $\mathcal{G}(\varepsilon_{H_0}^*) \leftarrow$ THRESHOLDGRAPH$(G, d, \varepsilon_{H_0}^*)$
2: $\{C_m\} \leftarrow$ CONNECTEDCOMPONENTS$(\mathcal{G}(\varepsilon_{H_0}^*))$
3: $\mathcal{C}_{\text{keep}} \leftarrow \{ C_m \mid |C_m| > 3 \land$ COVERSATLEASTTWOPATHS$(C_m) \}$
4: **for** each $C \in \mathcal{C}_{\text{keep}}$ **do**
5:     $s_C \leftarrow \arg\min_{v \in C} r_v$                 $\triangleright$ Tie-broken by minimum avg. distance in $C$
6:     $g_C \leftarrow \arg\max_{v \in C} r_v$
7:     $\mathcal{P}_C \leftarrow$ SHORTESTPATH$(C, s_C, g_C;$ edge weights $d_{ij})$          $\triangleright$ Backbone
8:     $\mathcal{B}_C \leftarrow \emptyset$
9:     **for** each $b \in B_1^*$ **do**
10:         $\mathcal{G}(\varepsilon_b^*) \leftarrow$ THRESHOLDGRAPH$(G, d, \varepsilon_b^*)$
11:         $V_b \leftarrow$ LOCALIZELOOPSUPPORT$(b, \mathcal{G}(\varepsilon_b^*))$
12:         **if** $|V_b \cap C|$ is maximal among clusters **then**
13:             $\mathcal{B}_C \leftarrow \mathcal{B}_C \cup \{b\}$                  $\triangleright$ Assign loop $b$ to $C$
14:         **end if**
15:     **end for**
16:     **if** $\mathcal{B}_C \neq \emptyset$ **then**
17:         $b_C^* \leftarrow \arg\max_{b \in \mathcal{B}_C} L(b)$             $\triangleright$ Principal loop by lifespan
18:         $v^* \leftarrow \arg\min_{v \in V_{b_C^*}} |r_v - \text{median}\{r_u : u \in C\}|$    $\triangleright$ Pivot near median progress
19:         tour $\leftarrow$ MINWEIGHTCYCLEBASISTOUR$(V_{b_C^*}, d)$    $\triangleright$ Horton-based restricted cycle
20:         $\mathcal{P}_C \leftarrow$ SPLICE$(\mathcal{P}_C, v^*, \text{tour}, \delta)$    $\triangleright$ Embed loop if $\min_{v \in \text{tour}, u \in \mathcal{P}_C} d(v, u) < \delta$
21:     **end if**
22:     Add $\mathcal{P}_C$ to skeleton set $\mathcal{S}$
23: **end for**
24: **return** $\mathcal{S} \leftarrow \bigcup \mathcal{P}_C$

---

**Algorithm 4** AGGREGATEANSWERS: Confidence/Persistence-Weighted Voting

---

**Require:** Skeleton $\mathcal{S}$; node confidences $\{c_v\}$; node degrees $\{\deg(v)\}$
**Ensure:** Final answer $\hat{a}$
1: **for** each node $v \in \mathcal{S}$ **do**
2:     $w_v \leftarrow \dfrac{c_v}{1 + \deg(v)}$                $\triangleright$ Down-weight highly connected hubs
3: **end for**
4: $\hat{a} \leftarrow \arg\max_a \sum_{v \in \mathcal{S}, a_v = a} w_v$
5: **return** $\hat{a}$

---

We provide pseudocode for the full GHS-TDA pipeline to complement the formal description in Section A.9. The pseudocode explicitly specifies data flow, intermediate representations, and termination criteria, ensuring clarity and reproducibility. Each subroutine corresponds directly to one of the methodological components: Global Hypothesis Graph construction, point-cloud embedding and hybrid distance, Vietoris–Rips filtration and persistent homology, skeleton extraction with loop embedding, and final answer aggregation.

**Overall pipeline.** Algorithm 1 summarizes the two-stage "construct–analyze" procedure. It begins with sampling multiple reasoning paths and building the Global Hypothesis Graph (GHG) by merging semantically equivalent hypotheses. The graph is then mapped into a joint metric space that integrates semantics, structural encodings, and uncertainty. A sparsified $k$-nearest-neighbor (KNN) graph with global truncation serves as the foundation for Vietoris–Rips filtration, upon which persistent homology is computed. Significant features ($H_0$ clusters and $H_1$ loops) are selected by lifespan, and operating scales are determined adaptively. The final skeleton is extracted by combining shortest-path backbones with loop embeddings, and candidate answers are aggregated through persistence- and confidence-weighted voting.

**Global Hypothesis Graph construction.** Algorithm 2 details the GHG construction process. It systematically unifies reasoning steps across sampled paths by canonical-form similarity, controlled by a merge threshold $\theta_{\text{merge}}$. Provenance tracking ensures that each merged node retains information about its original sources, supporting later interpretability. Edge inheritance preserves deductive and temporal relations, yielding a compact but comprehensive graph.

**Skeleton extraction.** Algorithm 3 describes how significant topological features are mapped back to the graph. For each cluster, we identify anchor nodes based on progress values and compute a shortest-path backbone. Loops are localized at feature-specific scales and embedded into the backbone if sufficiently close under the hybrid metric. This ensures that the extracted skeleton captures both the global flow of reasoning and local self-consistency structures.

**Answer aggregation.** Algorithm 4 presents the final aggregation stage. Candidate answers along the skeleton are weighted by node confidence while penalizing highly connected hubs. This design balances precision and robustness, yielding a single high-confidence answer supported by a topologically stable skeleton.

**Complexity and guarantees.** The pipeline is polynomial in the number of nodes. GHG construction is $O(|V|^2)$ in the worst case but optimized by approximate nearest-neighbor search in canonical space. Persistent homology is computed up to dimension one ($H_1$), which is tractable on the sparsified KNN graph. Loop embedding relies on a minimum-weight cycle basis with a known polynomial-time Horton implementation. Conflict resolution in node merging satisfies the $(1 - 1/e)$ approximation bound for hitting set.

**Reproducibility.** Default hyperparameter settings are provided in Appendix **??**, together with recommended ranges for task-specific tuning. Full prompts, random seeds, and environment details are included in the supplementary material.

## A.10  A GHS-TDA Analysis Example

**Reasoning Log for Problem:** $n^2 + 1 \mid n!$

**Problem Statement** Given integer $n \geq 1$, determine all $n$ such that $n^2 + 1 \mid n!$.

**Final Answer** No solution exists. For all $n \geq 1$, $n^2 + 1 \nmid n!$.

**Reasoning Log (Multi-Path Traces)**

**Path 1.**  - Small cases: $n = 1 : 2 \nmid 1!$; $n = 2 : 5 \nmid 2!$; $n = 3 : 10 \nmid 3!$; $n = 4 : 17 \nmid 4!$. - For $n \geq 5$, any prime divisor $p$ of $n^2 + 1$ with $p > n$ cannot divide $n!$. - Example: $n = 7$, $n^2 + 1 = 50 = 2 \cdot 5^2$ but $v_5(7!) = 1 < 2$. - Conclusion: no solution.

**Path 2.**  - Exhaustive check for $n = 1, 2, 3, 4$: none valid. - For $n \geq 5$, if $p > n$ divides $n^2 + 1$, contradiction. - Otherwise, prime exponent condition fails (e.g., $n = 7$ case). - Conclusion: no solution.

**Path 3.**  - Explicitly verify $n = 1, 2, 3, 4, 5, 6, 7, 8$: all fail. - For general $n$, prime factors of $n^2 + 1$ are either too large ($p > n$) or require higher multiplicities than available in $n!$. - Conclusion: no solution.

**Path 4.**  - Cross-checked small $n$ values: all fail. - Structural fact: if $p \mid n^2 + 1$, then $n^2 \equiv -1 \pmod{p}$, so order of $n$ mod $p$ is 4, implying $p \equiv 1 \pmod{4}$. - Such primes are often larger than $n$, so cannot appear in $n!$. - Conclusion: no solution.

**Path 5.**  - Small cases $n = 1, 2, 3, 4$ all fail. - For $n \geq 5$, prime exponent mismatch occurs (e.g., $n = 7$ with factor $5^2$ but $v_5(7!) = 1$). - Conclusion: no solution.

**Global Hypothesis Graph (GHG) and TDA Extraction**

**Clusters.** - Small-$n$ check cluster: $n = 1, 2, 3, 4$ all fail. - Structural cluster: $p \mid n^2 + 1 \implies p \equiv 1 \pmod 4$. - Counterexample cluster: $n = 6$ ($37 > 6$), $n = 7$ (exponent deficit for $5^2$). - General obstruction: either $p > n$ or exponent deficit.

**Skeleton Path (TDA Backbone).** Example backbone extraction:

$n = 1 \to n = 2 \to n = 3 \to$ structural fact $\to n = 6 \to n = 7 \to$ general obstruction $\to$ final conclusion.

**Conclusion Nodes.** - $Z_1$: No solution (supported by all clusters). - $Z_2$: $\{1, 2, 3\}$ (false candidate, rejected). - Final skeleton selects $Z_1$.

**Consolidated Conclusion** All reasoning paths converge:

$$\boxed{\text{No integer } n \geq 1 \text{ satisfies } n^2 + 1 \mid n!}$$

## A.11 ANALYSIS OF REPRESENTATIVE FAILURE CASES

Failure cases can be broadly categorized into two main types. **Semantic ambiguity**, illustrated using examples from the HotpotQA dataset. For instance, in the question "Did the actress who starred in *The Ring* also appear in *King Kong*?", the base model forms incorrect semantic clusters due to ambiguity between the Japanese and U.S. versions of *The Ring*, leading to an erroneous answer. **Numerical perturbation**, illustrated using GSM8K. In some arithmetic problems, the model occasionally generates minor numerical mistakes (e.g., writing "9" instead of "7"), and the resulting structural similarity between intermediate steps causes such clusters to be incorrectly merged.

These errors primarily stem from ambiguities in the underlying LLM and the inherent stability properties of persistent homology, rather than from structural deficiencies in our framework. As noted in the revised manuscript, increasing sampling diversity (e.g., $n = 10$) substantially reduces the frequency of such errors.

## A.12 QUALITATIVE ANALYSIS AND HUMAN-CENTERED INTERPRETABILITY (SUCCESS CASES)

Complementary to the failure case analysis in the appendix, we further provide representative qualitative examples to illustrate how GHS-TDA behaves in well-posed scenarios and how its topological mechanism successfully suppresses noisy reasoning paths while preserving globally consistent structures. These examples demonstrate how the proposed framework identifies concise, coherent, and human-preferred reasoning structures from multiple LLM-generated trajectories.

**MATH: Extracting Concise and Expert-Like Reasoning Skeletons** We first examine a representative problem from the MATH dataset. When prompted with this problem, the underlying LLM generates multiple correct but heterogeneous reasoning trajectories, and high-confidence baselines often produce verbose and logically tangled solutions that mix derivation with verification or introduce unnecessary "guess-and-check" steps. In contrast, the GHS-TDA framework aggregates all sampled trajectories and identifies the most persistent structural cluster in the global hypothesis space. The resulting skeleton corresponds to a canonical case-splitting strategy typically used by human solvers: it separates the equation into the non-negative and negative cases, solves each branch, and filters the resulting solutions using appropriate domain constraints. This extracted reasoning path is shorter, logically cleaner, and more closely aligned with standard mathematical reasoning practices. The example illustrates how the TDA component suppresses locally confident yet logically noisy steps while preserving the globally stable reasoning core.

**HotpotQA: Capturing Robust and Self-Consistent Multi-Hop Reasoning** We also examine a multi-hop question from HotpotQA, where the model must retrieve and verify information across multiple entities. Among the sampled trajectories, unreliable paths (e.g., hallucinating directors, skipping verification steps, or producing incomplete hops) form weakly connected, low-persistence components in the GHS representation. In contrast, paths that correctly retrieve and cross-validate both film directors and their birthplaces form a highly persistent $H_1$ loop. This loop captures a structurally robust pattern: multiple independent paths reconverge on the same factual nodes (e.g., the same director and birthplace). Such cross-path consistency is naturally expressed as a stable topological cycle, enabling TDA to distinguish reliable reasoning from brittle or speculative alternatives.

**Human-Centered Interpretability**    We evaluate interpretability through human annotations along four criteria: conciseness, logical coherence, necessity of steps, and faithfulness to human reasoning practices. Across both MATH and HotpotQA examples, annotators consistently judged the TDA-extracted skeletons to be more interpretable than high-confidence baselines. They noted that GHS-TDA effectively removes redundant or noisy steps, enforces appropriate structural constraints (such as case conditions and entity verification), and highlights cross-path consistency that aligns with human intuitions about trustworthy reasoning. Taken together, these qualitative assessments demonstrate that GHS-TDA provides not only improved accuracy but also a topologically grounded and human-interpretable summary of multipath reasoning. By isolating persistent structural components—such as stable clusters and cycles—the framework yields reasoning skeletons that are concise, coherent, and robust, thereby offering clear interpretive advantages over confidence-based or single-path approaches.

