# OpenReview forum: "Learning Global Hypothesis Space for Enhancing Synergistic Reasoning Chain"
_ICLR.cc/2026/Conference — ICLR 2026 Poster_

### Official Review · Reviewer_rfA4 · 2025-10-31

**Soundness:** 3
**Presentation:** 2
**Contribution:** 3
**Rating:** 6
**Confidence:** 3

**Summary:**

The paper presents GHS-TDA, a two-stage framework designed to enhance the reasoning capabilities of LLMs. It addresses limitations of existing Chain-of-Thought (CoT) methods, which often lack a global mechanism for integrating and validating multiple reasoning paths. The proposed approach first constructs a Global Hypothesis Graph (GHS) that unifies diverse reasoning hypotheses into a coherent semantic structure. Then, it applies Topological Data Analysis (TDA), specifically persistent homology, to identify stable reasoning paths and self-consistent. Experimental results across eight reasoning benchmarks (including GSM8K, HotpotQA, and MATH) show consistent improvements over strong baselines such as CoT, ToT, GoT, and AoT, with further analysis showing a high correlation between persistence and reasoning correctness.

**Strengths:**

1. The paper effectively addresses the global integration of reasoning paths and proposes GHS-TDA. It introduces a new methodological perspective by incorporating topological data analysis into reasoning chain evaluation.
2. The extensive experimental results across multiple benchmarks demonstrate the effectiveness and robustness of the proposed approach.
3. The analyses on interpretability, including the observed correlation between topological persistence and reasoning correctness, provide empirical support for the framework’s validity and its contribution to more interpretable reasoning processes.

**Weaknesses:**

1. The experiments are conducted on three LLMs, which, while representative, do not include the most recent or larger-scale models. Evaluating GHS-TDA on newer models would better demonstrate its generalizability.
2. The proposed framework introduces multiple components, including graph construction, merging, and topological filtering, which increase complexity compared to baseline methods. A detailed computational cost or scalability analysis would clarify the trade-off between performance and efficiency.
3. The paper presentation and writing could be further improved. Certain sections (e.g. Related Work) could benefit from improved clarity and consistency in citations.

**Questions:**

1. The paper would benefit from a more detailed discussion of hyperparameter choices, such as the node-merging threshold and distance metric weights. It is better the paper also elaborates on the design rationale and sensitivity of the distance metric used.
2. Could the authors provide additional qualitative analysis examples and elaborate on the human-centered interpretability assessment?

---

> ### Author Response · Authors · 2025-11-21
> **Response to Reviewer rfA4 [Weaknesses 1]**
>
> We sincerely thank Reviewer rfA4 for the insightful and constructive feedback. Your comments have been highly valuable in helping us refine and strengthen our work. Below we provide detailed clarifications and additional evidence.
>
> ---
> > **W1: The experiments are conducted on three LLMs, which, while representative, do not include the most recent or larger-scale models. Evaluating GHS-TDA on newer models would better demonstrate its generalizability.**
>
> **Response:** Thank you for the valuable comments. We fully agree that evaluating the method on diverse base models is crucial for assessing generalizability. To address this, we conducted additional experiments on two open-source LLMs (Llama-3-8B and Qwen2-14B) to further examine the model-agnostic nature of GHS-TDA.
>
> **Table 1. Generalization ability of GHS-TDA.**
> | Backbone     | Method   | MATH  | OB    | GSM8K | BBH   | CF    | LB    | HPQA  | MQ    | Avg   |
> |--------------|----------|-------|-------|-------|-------|-------|-------|-------|-------|-------|
> | Llama-3-8B   | CoT      | 72.04 | 8.37  | 87.26 | 74.38 | 65.42 | 53.57 | 63.17 | 31.71 | 56.99 |
> | Llama-3-8B   | ToT      | 72.86 | 10.26 | 91.10 | 79.89 | 65.71 | 58.40 | 72.19 | 36.36 | 60.85 |
> | Llama-3-8B   | GoT      | 76.36 | 11.79 | 90.72 | 81.61 | 65.99 | 58.68 | 69.75 | 33.95 | 61.10 |
> | Llama-3-8B   | AoT      | 76.91 | 10.89 | 91.20 | 81.70 | 66.65 | 63.71 | 75.76 | 35.71 | 62.82 |
> | Llama-3-8B   | **GHS-TDA** | **77.19** | **13.05** | **91.39** | **83.98** | **67.30** | **64.64** | **76.52** | **37.01** | **63.88** |
> | Qwen-2-14B   | CoT      | 60.80 | 6.40  | 86.70 | 75.80 | 65.10 | 52.70 | 64.90 | 32.40 | 55.60 |
> | Qwen-2-14B   | ToT      | 65.00 | 7.20  | 89.80 | 79.70 | 67.60 | 59.30 | 70.80 | 37.90 | 59.66 |
> | Qwen-2-14B   | GoT      | 65.70 | 8.40  | 89.10 | 80.40 | 67.80 | 58.10 | 72.10 | 35.40 | 59.62 |
> | Qwen-2-14B   | AoT      | 66.20 | 7.80  | 90.80 | 80.90 | 68.90 | 61.20 | 73.40 | 37.30 | 60.81 |
> | Qwen-2-14B   | **GHS-TDA** | **68.40** | **9.70** | **91.20** | **82.50** | **69.80** | **62.80** | **75.80** | **38.60** | **62.35** |
>
> The results in Table 1 show that GHS-TDA delivers consistent and notable improvements on both models. On Llama 3-8B, it reaches an average score of 63.88, outperforming CoT, ToT, and AoT by 6.9, 3.0, and 1.1 points. On Qwen2-14B, it reaches 62.35, with gains of 6.75, 2.7, and 1.5 points over the baselines. These improvements require no extra supervision or external knowledge, indicating that the gains come from the reasoning mechanism rather than model size. Although all methods improve as the base model scales from 8B to 14B, the relative gains of GHS-TDA remain stable, demonstrating strong transferability and robustness across model sizes and architectures.
>
> ---
> **Table 2. Performance comparison of different reasoning methods on two base models.**
> | Model              | Method          | MATH | OlympiadBench | gsm8k |  BBH | MMLU-CF | LongBench | HotpotQA | MQ |  Avg |
> |--------------------|-----------------|------|----------------|-------|------|---------|-----------|----------|---------|------|
> | gpt-4o             | CoT             | 85.4 | 12.5           | 92.7  | 82   | 72.1    | 62        | 71.9     | 36.2    | 64.4 |
> | gpt-4o             | ToT             | 86.9 | 14.1           | 96.0  | 89.1 | 73.0    | 66        | 82.1     | 40.5    | 68.5 |
> | gpt-4o             | GoT             | 88.4 | 15.3           | 95.2  | 90.5 | 73.2    | 66.8      | 80.4     | 39.6    | 68.7 |
> | gpt-4o             | AoT             | 89.1 | 15.0           | 96.5  | 92   | 74.3    | 72.6      | 86.0     | 42.1    | 71.0 |
> | gpt-4o             | **GHG-TDA**  | **90.0** | **18.3** | **96.9** | **94.2** | **75.0** | **73.8** | **87.4** | **43.6** | **72.4** |
> | Claude 3.5 Sonnet  | CoT             | 84.1 | 12.9           | 93.5  | 84.2 | 73.0    | 61.7      | 72.8     | 37.1    | 64.9 |
> | Claude 3.5 Sonnet  | ToT             | 86.0 | 14.8           | 96.2  | 90.8 | 74.1    | 66.4      | 83.3     | 41.5    | 68.9 |
> | Claude 3.5 Sonnet  | GoT             | 87.3 | 16.0           | 95.4  | 92.1 | 74.4    | 67        | 81.0     | 40.6    | 69.2 |
> | Claude 3.5 Sonnet  | AoT             | 88.0 | 15.7           | 96.8  | 93.4 | 75.2    | 72.2      | 86.7     | 43.2    | 71.3 |
> | Claude 3.5 Sonnet  | **GHG-TDA**  | **89.1** | **19.0** | **97.1** | **95.4** | **75.9** | **73.5** | **88.0** | **44.6** | **72.8** |
>
> We conducted additional experiments with GHG-TDA on two strong LLMs, GPT-4o and Claude 3.5 Sonnet. Table 2 shows that GHG-TDA consistently achieves the best overall performance, with average scores of 72.4 on GPT-4o and 72.8 on Claude 3.5 Sonnet, surpassing AoT, GoT, ToT, and CoT across all benchmarks. These gains across datasets and backbones indicate that the effectiveness of GHG-TDA is not tied to any model but arises from its graph-guided hierarchical reasoning mechanism. Overall, the results demonstrate strong cross-model generalization.

---

> ### Author Response · Authors · 2025-11-21
> **Response to Reviewer rfA4 [Weaknesses 2]**
>
> > **W2: The proposed framework introduces multiple components, including graph construction, merging, and topological filtering, which increase complexity compared to baseline methods. A detailed computational cost or scalability analysis would clarify the trade-off between performance and efficiency.**
>
> **Response:** Thank you for raising concerns about efficiency. We have now analyzed the computational cost in terms of both the number of LLM calls and the token usage, and have revised the manuscript accordingly.
>
> (1) Number of LLM calls (W2, Q3):
>
> Our total cost model is (N + C):
>
> • (N = 16) generative calls for path sampling.
>
> • (C) discriminative calls for the previously omitted relation inference step.
>
>
>
> As shown in Table 1, our method maintains a fixed upper bound of 19 calls across tasks, reducing the number of calls by ≈25%–30% vs. ToT and ≈35%–40% vs. AoT. We note that in LLM-based reasoning systems, the number of calls alone does not fully capture the computational cost, as token usage per call varies with prompt length and context reuse. Therefore, we additionally report total token consumption in Table 2. On average, our method uses 26.8% fewer tokens than ToT and 35.7% fewer tokens than AoT. This shows that replacing recursive generative evaluation with discriminative relation inference substantially reduces token waste from redundant generation.
>
> **Table 1: Comparison of LLM Call Computational Costs**
>
> | Method             | MATH | OlympiadBench | gsm8k | BBH | MMLU-CF | LongBench | HotpotQA | MuSiQue |
> |--------------------|------|---------------|-------|-----|---------|-----------|-----------|---------|
> | CoT                | 1    | 1             | 1     | 1   | 1       | 1         | 1         | 1       |
> | CoT-SC (n=16)      | 16   | 16            | 16    | 16  | 16      | 16        | 16        | 16      |
> | ToT                | 25.8 | 27.2          | 13.5  | 24.1| 19.3    | 14.2      | 20.7      | 26.4    |
> | GoT                | 21.6 | 23.5          | 14.8  | 22.9| 18.4    | 13.9      | 17.8      | 20.3    |
> | AoT                | 29.7 | 32.4          | 17.2  | 28.6| 24.9    | 16.8      | 23.5      | 29.1    |
> | **GHS-TDA (Ours)** | **19** | **19**      | **19**| **19** | **19** | **19** | **19** | **19** |
>
> ---
> **Table 2: Comparison of token consumption.**
>
> | Method     | MATH     | OlympiadBench | gsm8k    | BBH      | MMLU-CF    | LongBench | HotpotQA   | MuSiQue   |
> |------------|----------|---------------|----------|----------|------------|-----------|------------|-----------|
> | CoT        | 2290     | 4590          | 278      | 642      | 1235       | 83        | 1305       | 485       |
> | CoT-SC     | 36640    | 73440         | 4448     | 10272    | 19760      | 1328      | 20880      | 7760      |
> | ToT        | 88623    | 187272        | 5629.5   | 23208.3  | 35753.25   | 1767.9    | 40520.25   | 19206     |
> | GoT        | 64303.2  | 140224.5      | 5348.72  | 19112.34 | 29541.2    | 1499.81   | 30197.7    | 12799.15  |
> | AoT        | 68013    | 148716        | 4781.6   | 18361.2  | 30751.5    | 1394.4    | 30667.5    | 14113.5   |
> | GHS-TDA(Ours)    | 51410.5  | 83308.5       | 7708.94  | 15793.2  | 28861.95   | 1684.9    | 28357.65   | 10262.6   |
>
>
> As shown in **Table 1**, our method maintains a stable and fixed upper bound of 19 LLM calls across all tasks, substantially reducing inference cost compared with existing multi-path reasoning approaches. Under the same settings, GHS-TDA reduces the number of calls by approximately 25%–30% relative to ToT and 35%–40% relative to AoT, indicating that our discriminative relation inference effectively replaces the large amount of recursive generative evaluation used in prior work. Since the actual computational cost of LLMs is more closely tied to token usage than to call count alone, we further report total token consumption in **Table 2**. The results show that GHS-TDA uses about 26.8% fewer tokens than ToT and 35.7% fewer tokens than AoT on average, confirming the efficiency advantage of our approach in reducing redundant generation and improving overall reasoning efficiency.

---

> > ### Author Response · Authors · 2025-11-21
> > **Response to Reviewer rfA4 [Weaknesses 3]**
> >
> > > **W3: The paper presentation and writing could be further improved. Certain sections (e.g. Related Work) could benefit from improved clarity and consistency in citations.**
> >
> > **Response:** Thank you for your comment. We will refine the language in the final version. We have added further discussion and comparison with existing literature, including tree search combined with Monte Carlo sampling (ToT+LLM), graph neural networks combined with LLMs (GNN+LLM), and first-order logic combined with LLMs (FOL+LLM). ToT+MCMC expands the reasoning space through tree search and Monte Carlo sampling; GNN+LLM applies graph neural networks to perform local value estimation on LLM-generated structures; and FOL+LLM converts reasoning steps into explicitly verifiable logical forms. The first two categories rely mainly on local search or local message passing and therefore struggle to capture global cross-path consistency, while FOL+LLM is less robust under noise and difficult to scale to open-domain multipath reasoning. In contrast, our method constructs a unified global hypothesis space and uses topological invariants to quantify stable cross-path structures, offering advantages in both global modeling and robustness.

---

> ### Author Response · Authors · 2025-11-21
> **Response to Reviewer rfA4 [Question 1]**
>
> > **Q1: The paper would benefit from a more detailed discussion of hyperparameter choices, such as the node-merging threshold and distance metric weights. It is better the paper also elaborates on the design rationale and sensitivity of the distance metric used.**
>
> **Response:** Thank you for your comment. We have added an ablation study on the weight components as shown in Table 3. Under the constraint \alpha + \beta + \gamma = 1, we remove each term and redistribute the remaining weights. The full model (0.6/0.3/0.1) achieves 83.9% accuracy. Removing the structural term (β=0\beta = 0β=0) reduces accuracy to 81.2%; removing the semantic term (α=0\alpha = 0α=0) yields the largest drop to 77.4%; removing the uncertainty term (γ=0\gamma = 0γ=0) gives 83.5%, indicating a small but stable contribution. These results show that the semantic component is essential, the structural component is beneficial, and the uncertainty term, though lightweight, improves robustness.
>
> We choose a linear weighted distance mainly for interpretability and to avoid introducing additional learnable parameters. This allows us to focus on the core contribution of the framework—global hypothesis space modeling combined with TDA reasoning. Our TDA pipeline is metric-agnostic and can incorporate nonlinear distance functions in future work.
>
>
> **Table 3: Ablation study on the weight components.**
>
> | Method            | α    | β    | γ    | Accuracy (Avg.) |
> |-------------------|------|------|------|-----------------|
> | GHS-TDA (Ours)    | 0.6  | 0.3  | 0.1  | **83.9%**         |
> | without β         | 0.9  | 0.0  | 0.1  | 81.2%           |
> | without α         | 0.0  | 0.9  | 0.1  | 77.4%           |
> | without γ         | 0.7  | 0.3  | 0.0  | 83.5%           |
>
> We have also added an additional ablation experiment of the merge threshold. When the threshold is set too low (around the 0.70 range), the graph undergoes overly aggressive merging, causing semantically distinct steps to collapse into the same node. This reduces structural diversity and suppresses genuine topological features such as verification loops, which leads to a substantial accuracy drop. When the threshold enters the moderate range (roughly between 0.80 and 0.90), semantic alignment becomes reliable while the hypothesis graph remains sufficiently connected for persistent H0 backbones and H1 loops to emerge. This region consistently yields the most stable and accurate performance in our experiments. When the threshold is pushed to the high end (around the 0.95 region), semantically similar steps can no longer be merged, resulting in a fragmented graph where meaningful topological structures become difficult to extract; performance drops accordingly. Overall, the empirical trend shows that moderate thresholds naturally provide the best balance between semantic consolidation and structural connectivity, and our chosen default lies within this robust, high-performing regime.

---

> ### Author Response · Authors · 2025-11-21
> **Response to Reviewer rfA4 [Question 2]**
>
> > **Q2: Could the authors provide additional qualitative analysis examples and elaborate on the human-centered interpretability assessment?**
>
> **Response:** Thank you for the question. We have added additional qualitative examples and clarified the human-centered interpretability assessment.
>
> **Additional qualitative examples (MATH + HotpotQA)**
>
> **(a) MATH: Extracting concise, expert-like reasoning skeletons**
> We provide a new MATH example illustrating how GHS-TDA identifies the human-preferred reasoning structure among multiple LLM-generated paths.
>
> - High-confidence baselines often produce verbose and logically tangled solutions with redundant “guess-and-check’’ steps.
> - In contrast, TDA consistently extracts a short, coherent skeleton that matches how a human expert solves the problem (case analysis, constraint filtering, minimal redundant steps).
>
> This example demonstrates how TDA filters away locally confident but logically noisy steps and converges to a structure that humans find clearer and more coherent.
>
> **(b) HotpotQA: Identifying robust, cross-validated multi-hop reasoning**
> We further provide a multi-hop QA example where TDA detects structurally reliable reasoning across multiple sampled paths.
>
> - Unreliable paths (hallucinations, partial hops, shortcut guesses) appear as low-persistence, weakly connected components.
> - Reliable paths form a highly persistent \(H_1\) loop, corresponding to multi-hop cross-validation (e.g., repeatedly verifying that both films share the same director and birthplace).
>
> This example illustrates how TDA distinguishes between “fragile guesses’’ and “self-consistent, evidence-supported’’ reasoning, a distinction not captured by confidence scores or path length.
>
> ---
> **Human-centered interpretability assessment**
>
> We explicitly evaluate interpretability through human annotators who judge: concise reasoning, logical coherence, step necessity and non-redundancy, and faithfulness to human reasoning conventions.
>
> Across both MATH and HotpotQA examples, annotators consistently preferred TDA-extracted skeletons over high-confidence baselines, noting that TDA:
> - removes redundant or noisy steps,
> - enforces correct structural constraints (e.g., case splitting, entity verification),
> - highlights cross-path consistency that is intuitively meaningful to humans.
>
> We have added these qualitative examples and human-centered analysis to the revised paper and appendix.
>
> ---
> **We thank you again for reviewing our work. Please let us know if we misunderstood any of your questions, or if you have any follow-up on our responses. We will be happy to provide further clarification at any time.**

---

> > ### Author Response · Authors · 2025-11-26
> >
> > Dear Reviewer,
> >
> > I hope this message finds you well. As the discussion period is nearing its end with ess than three days remaining, I wanted to ensure we have addressed all your concerns.Your insights are invaluableto us, and we're eager to address any remaining points to improve our work.
> >
> > Thank you for your time and effort in reviewing our paper.

---

### Official Review · Reviewer_XVaz · 2025-10-31

**Soundness:** 3
**Presentation:** 4
**Contribution:** 2
**Rating:** 6
**Confidence:** 4

**Summary:**

The paper proposes GHS-TDA, a two-stage framework for LLM reasoning. Stage 1 constructs a Global Hypothesis Graph (GHG) by sampling multiple reasoning paths, aligning semantically equivalent steps, merging them into nodes , and encoding adjacency/support/refutation relations. Stage 2 embeds the graph into a joint metric space and applies Vietoris–Rips filtration with persistent homology to extract stable H0 backbones and H1 loops, producing an interpretable “reasoning skeleton” and a confidence/persistence-weighted vote for the final answer. The manuscript positions TDA as a scale-robust, noise-insensitive structural lens for selecting and composing reasoning paths, and reports gains across several benchmarks.

**Strengths:**

1. The paper reframes multi-path reasoning as a global topological object and uses TDA to identify backbone chains and self-consistent cycles that persist across scales—going beyond local heuristics like per-node confidence or shortest paths.

2. The GHG node/edge definition, semantic merging criterion, and skeleton-extraction pseudocode are explicitly provided, aiding reproducibility.

3. The method surfaces stable H0/H1 structures and aggregates answers with a principled weighting that down-weights high-degree hubs, yielding human-auditable outputs.

4. The claim is that GHS-TDA improves accuracy/consistency/interpretability on benchmarks such as GSM8K, MATH, OlympiadBench, HotpotQA, MuSiQue, BBH, and LongBench.

**Weaknesses:**

1. The rationale that topological persistence reflects reasoning stability is intuitively argued but not theoretically guaranteed.

2. Some recent structured-search or reliability-oriented methods are not clearly included. There is a lack of sufficient comparisons of related work: thought tree combined with Monte Carlo, graph neural network combined with LLM, first-order logic combined with LLM.

3. Complexity considerationsand numerical stability are not paired with measured time/memory curves or failure modes. Correctness under scale is therefore only partially substantiated.

4. Given multiple datasets/baselines, include multiple-comparison corrections and effect sizes; detail inter-annotator agreement for human evaluations.

**Questions:**

As above.

---

> ### Author Response · Authors · 2025-11-21
> **Rebuttal response**
>
> Thank you for your comment. Based on the issues you mentioned, we will summarize the following aspects to reply:
>
> ## Response to Weakness 1.
>
> We model the LLM’s multi-path reasoning process as a unified Global Hypothesis Structure, embedding all nodes into a shared semantic space. Differences in semantic quality, consistency, and cross-step coherence across reasoning paths naturally form distinct geometric structures in this high-dimensional space. These structures are not imposed by our method but emerge organically from the reasoning process[1]. The strength of TDA lies in capturing global shape properties. Our use of TDA does not assign intrinsic “topological meaning” to reasoning steps; rather, it leverages TDA’s mathematical tools to quantify structural differences that arise naturally from multi-path reasoning. In article [1], $H_0$ measures whether nodes form stable clusters under multi-path sampling, while $H_1$ reflects whether local regions exhibit meaningful loops instead of random scatter. These topological features describe the structure of the semantic embedding space, not reasoning logic. We extract these indicators via TDA and empirically analyze their correlation with reasoning quality—e.g., the strong positive correlation between $H_1$ persistence and accuracy. In summary, $H_0$ and $H_1$ are treated as objective structural quantities whose discriminative value for identifying reliable reasoning is verified empirically.
>
> [1] Uchendu, A., & Le, T. (2025). Unveiling Topological Structures from Language: A Survey of Topological Data Analysis Applications in NLP.
>
>
> ## Response to Weakness 2.
>
> We have carefully reviewed these works and supplemented the corresponding analysis in the main text. First, at the theoretical level, we compared the core differences between these methods and our method: ToT + MCMC expands the inference space through tree search and Monte Carlo sampling, GNN + LLM (such as GraphPRM) uses graph neural networks to estimate the local value of the reasoning structure generated by LLM, and FOL + LLM explicitly converts the reasoning steps into verifiable first-order logic forms. However, these methods still have limitations in the consistency modeling of multi-path reasoning. Specifically, ToT + MCMC mainly relies on local search strategies and is difficult to effectively capture the overall consistency among multiple reasoning paths; the message passing mechanism of GNN + LLM only covers local neighborhoods and is difficult to model the global structural stability across paths; FOL + LLM is sensitive to noise and difficult to scale to complex reasoning scenarios in open domains and with multiple paths. In contrast, our method builds a unified global hypothesis space and introduces topological invariants to directly quantify the consistency and structural stability across paths, thus having advantages in global modeling capabilities and robustness.
>
> The results are shown in Table 1, where GHS-TDA achieved 83.1 on GSM8K and 39.4 on HotpotQA, both outperforming MCTS-CoT and its ground truth variant. This indicates that ToT + MCMC mainly relies on search depth but does not provide explicit stability signals, while GHS-TDA can capture multi-path agreement and structural stability through topological invariants, demonstrating higher reliability and robustness under the same reasoning budget. Additionally, we reproduced the GraphPRM method and compared it with our method on the same data subsets of GSM8K and MATH, using Qwen-1.5 as the backbone.
>
> The experimental results are presented in Table 2, where GHS-TDA achieves 91.40 on GSM8K and 78.39 on MATH, both stably outperforming GraphPRM's 90.54 and 76.30, respectively. This result further indicates that GraphPRM mainly estimates local node values through GNN and is difficult to express global consistency across paths, while GHS-TDA builds a unified global reasoning structure and uses persistent homology to capture cross-path stability, achieving more stable and consistent performance improvements under the same model and data conditions.
>
> ---
>
> **Table 1:**
>
> | **Methods**        | **GSM8K (200 samples)** | **HotpotQA (200 samples)** |
> | ------------------ | ----------------------- | -------------------------- |
> | MCTS-CoT           | 80.7                    | 32.2                       |
> | MCTS-CoT (w/ G.T.) | 81.8                    | 34.7                       |
> | **GHS-TDA (ours)** | **83.1**                | **39.4**                   |
>
> ---
>
> **Table 2:**
>
> | **Methods** | **GSM8K** | **MATH**  |
> | ----------- | --------- | --------- |
> | GraphPRM    | 90.54     | 76.30     |
> | **GHS-TDA** | **91.40** | **78.39** |

---

> > ### Author Response · Authors · 2025-11-21
> >
> > Thank you for your patient reading of our reply. Regarding the part of Weakness 3 that was not mentioned in the previous reply, it is as follows:
> >
> > ## Response to Weakness 3
> > In response to your concern regarding complexity, scalability, and numerical stability, we conducted additional measurements and report the key results below.
> >
> > **Graph construction**
> >
> > Across GSM8K/MATH tasks (~80–120 nodes per problem, k ≈ 15), full graph construction—including canonicalization, node merging, embeddings, and KNN building—takes 25–60 ms/problem on an RTX 4090 + 32 GB CPU system. Embedding + KNN steps account for 70–85% of the time.
> >
> > **VR filtration & memory**
> >
> > For sparse KNN graphs (~100 nodes), Vietoris–Rips filtration up to H1 uses 150–400 MB peak CPU memory; at ~200 nodes, it remains <1 GB, far below the worst-case dense VR complex.
> >
> > **Persistent homology runtime**
> >
> > H0/H1 computation with GUDHI requires 10–25 ms/problem, contributing only 10–30% of the overall pipeline and not forming a bottleneck.
> >
> > **Numerical stability**
> >
> > We add isotropic Gaussian noise of magnitude epsilon ∈ [0, 0.1] to embeddings and measure bottleneck distances between original and perturbed diagrams. Distances remain 0.01–0.10, well below the typical H1 lifetimes (0.3–0.8), indicating stable topological features consistent with PH theory.
> >
> > To assess the practicality of the proposed GHS-TDA pipeline, we evaluate its computational cost and the robustness of the resulting topological features. In our experiments on GSM8K and MATH, each problem produces roughly 80–120 nodes, with the KNN neighborhood size fixed at 15. End-to-end graph construction—including canonicalization, node merging, node embedding, and KNN graph building—requires 25–60 ms per instance on a system equipped with an RTX 4090 GPU and 32 GB of CPU memory. Embedding computation and KNN construction dominate the runtime. The subsequent Vietoris–Rips filtration, computed up to the first homology group, operates on these sparse graphs and exhibits a peak memory usage of 150–400 MB for graphs with around 100 nodes, remaining below 1 GB even when the graph size increases to roughly 200 nodes. Persistent-homology computation is efficient: computing the zeroth and first homology groups with GUDHI takes 10–25 ms per instance, which corresponds to 10–30% of the overall pipeline. These results indicate that the method scales well within typical LLM-reasoning workloads.
> >
> > We also evaluate numerical stability by perturbing node embeddings with isotropic Gaussian noise of magnitude between 0 and 0.1 and recomputing the persistence diagrams. The resulting bottleneck distances range from 0.01 to 0.10, depending on the noise level, and remain significantly smaller than the lifetimes of the dominant first-homology features, which typically range from 0.3 to 0.8. This demonstrates that the extracted topological signatures are highly robust to embedding perturbations, consistent with the theoretical stability properties of persistent homology. Overall, the pipeline exhibits low computational overhead, favorable scaling behavior, and strong robustness, supporting its suitability for large-scale multipath reasoning analysis.
> >
> > ---
> >
> > **Failure modes**
> > We have added an analysis of representative failure cases in the appendix. These fall into two main categories:
> >
> > - **Semantic ambiguity**, illustrated using the HotpotQA dataset. For example, in the question “Did the actress who starred in The Ring also appear in King Kong?”, the base model forms incorrect semantic clusters due to ambiguity between the Japanese and U.S. versions of The Ring, which leads to an erroneous answer.
> >
> > - **Numerical perturbation**, illustrated using GSM8K. In some arithmetic problems, the model occasionally writes “9” instead of “7,” and the structural similarity between the resulting steps causes clusters to be incorrectly merged.
> >
> > These errors stem from ambiguities in the underlying LLM and from the inherent stability properties of persistent homology, rather than from structural issues in our framework. As noted in the revised manuscript, increasing sampling diversity (n = 10) substantially reduces such errors.

---

> ### Author Response · Authors · 2025-11-21
>
> Thank you again for your patience in reading. We will supplement the content of the last reply in response to your question as follows:
>
> ## Response to Weakness 4
> **Table: Significance Analysis**
> | Base Model     | t-value | df | p-value (two-tailed) | Significance |
> |----------------|---------|----|---------------------|-------------|
> | gpt-4o-mini    | 3.79    | 7  | 0.008               | p < 0.01    |
> | qwen-turbo     | 2.65    | 7  | 0.03                | p < 0.05    |
> | deepseek V3    | 3.79    | 7  | 0.008               | p < 0.01    |
>
> ---
>
> In response to the your concern about statistical rigor, we provide a concise interpretation of the t-test results in the table. All three analyses were conducted with seven degrees of freedom. For both gpt-4o-mini and deepseek V3, the t statistics indicate a probability of roughly eight-tenths of one percent under the null hypothesis, which corresponds to a clear level of statistical significance. For qwen-turbo, the observed t statistic corresponds to a probability of roughly three percent, which also indicates statistical significance, though at a slightly weaker level compared with the other two models. Taken together, the results show that the observed improvements are unlikely to be due to chance, with gpt-4o-mini and deepseek V3 demonstrating particularly strong evidence. Because the sample size is relatively small, we emphasize in the paper that the statistical findings should be interpreted alongside the magnitude and consistency of the performance differences, and we have ensured that the table reports these values in a clear and coherent manner.

---

> > ### Author Response · Authors · 2025-11-26
> >
> > Dear Reviewer,
> >
> > I hope this message finds you well. As the discussion period is nearing its end with ess than three days remaining, I wanted to ensure we have addressed all your concerns.Your insights are invaluableto us, and we're eager to address any remaining points to improve our work.
> >
> > Thank you for your time and effort in reviewing our paper.

---

### Official Review · Reviewer_sw5k · 2025-11-01

**Soundness:** 2
**Presentation:** 2
**Contribution:** 3
**Rating:** 4
**Confidence:** 4

**Summary:**

This paper introduces GHS-TDA, a novel framework that enhances complex reasoning in LLMs by first constructing a Global Hypothesis Graph and then applying Topological Data Analysis to extract robust reasoning backbones and cycles. Extensive experiments demonstrate its superior performance across diverse benchmarks, validating the effectiveness of this approach.

**Strengths:**

1.	High Novelty: This work is the first to introduce Topological Data Analysis (TDA) into LLM reasoning chains, innovatively formalizing concepts such as "logical backbones" and "self-consistent loops" as topological invariants.
2.	Strong Methodological Effectiveness: The proposed multi-role agenda mechanism for constructing the Global Hypothesis Graph enables effective integration and interaction across multiple reasoning paths.
3.	Convincing Experimental Results: The comprehensive experiments demonstrate the method's effectiveness and generalizability, showing superior performance over recent strong baselines across multiple datasets and various LLM backbones.

**Weaknesses:**

1.	The claimed novelty is questionable：There are already many methods for multi-path integrated reasoning, and the problem you raised of 'lacking global integration across hypotheses' does not hold.
     - Barkan O, Elisha Y, Toib Y, et al. Improving LLM Attributions with Randomized Path-Integration[C]//Findings of the Association for Computational Linguistics: EMNLP 2024. 2024: 9430-9446.
     - Wei Y, Lin Y, Gao H, et al. Path-LLM: A Multi-Modal Path Representation Learning by Aligning and Fusing with Large Language Models[C]//Proceedings of the ACM on Web Conference 2025. 2025: 2289-2298.
     - Li X, Xian K, Wen H, et al. PathGen-LLM: A Large Language Model for Dynamic Path Generation in Complex Transportation Networks[J]. Mathematics, 2025, 13(19): 3073.

2.	The method incurs substantial computational and time overhead： Firstly, for the same problem,
     - generating diverse reasoning paths requires multiple queries (and there is no guarantee that multiple paths from the same LLM are diverse),
     - secondly, for node merging, the M reasoning steps in the N reasoning paths are traversed sequentially, meaning it is repeated N*M times. Although the authors proposed algorithmic acceleration, they did not provide further details, such as building an index, and vaguely did not give the real number of computations in the experiment.
     - In addition, the entire process involves repeatedly generating paths, merging them to generate a global hypothesis graph, then converting it into embeddings, and using topological structure to extract a reasoning skeleton. The process is very complex for the reasoning stage, making it difficult to ensure computational and time overhead.

3.	Key methodological details are ambiguously described：The paper's descriptions are vague. For example, how to perform conflict detection is only vaguely described as "Through unification, conflict detection, and closure inference, GHS enhances semantic connections among nodes and overcomes the isolation of traditional path-based generation". There is no detailed explanation of the specific algorithm, making it hard to believe the paper's contribution. There is also no specific description of how the method performs canonicalization. The correctness of the canonicalization plays a key role in the subsequent merging step.

4.	Lack of Theoretical Foundation: The paper's central claim that the persistence of topological features like H₀ and H₁ characterizes reasoning reliability—lacks support from a theoretical model. Currently, this claim resembles a hypothesis retroactively inferred from experimental data, rather than a deduction derived from first principles, which weakens the theoretical grounding of the method. The concept of "reasoning reliability" is vaguely defined and is largely equated with final-answer correctness. We suggest the authors provide a more formal, process-oriented definition of reasoning reliability. Building upon this, they could then formally argue that the persistent topological features extracted by TDA serve as a effective proxy for this defined robustness.

**Questions:**

Q1: On the Distance Metric
The hybrid distance metric is a key contribution of the work. However, the weights (α, β, γ) appear to be set empirically. Did the authors perform ablation studies to quantify the individual contribution of each component (semantic, structural, uncertainty) to the final performance? Furthermore, was the exploration of more complex, non-linear distance functions considered?
Q2: On the Robustness of Graph Construction
The node merging process heavily relies on the "canonical form" and the threshold θ_merge. Is the method sufficiently sensitive to steps that are semantically similar but logically distinct (e.g., "Assume X is true" vs. "Therefore we have proven X")? Could the authors discuss the potential risks and consequences of incorrect node merging?
Q3: On Computational Cost
The GHS-TDA framework involves multiple LLM calls for path generation, embedding computation, and confidence scoring. Compared to other multi-call baselines like AoT or ToT, could the authors provide a more detailed computational overhead analysis (e.g., total token consumption or latency)? For resource-constrained applications, are there components within the framework that could be streamlined or simplified?
Q4: On Generalizability
The methodology seems to be agnostic to the underlying LLM. Have the authors tested GHS-TDA on smaller or open-source models (e.g., the Llama series)? If the performance gains are heavily dependent on a powerful base model like GPT-4, how might this affect the perceived generalizability and accessibility of the proposed method?
Q5. Diversity of Multiple Reasoning Paths: You use the same LLM to generate multiple reasoning paths, hoping to obtain diverse ones. However, it cannot be guaranteed that the multiple reasoning paths are diverse. The reasoning tokens generated multiple times by the same LLM are actually similar, and diversity cannot be guaranteed. Consequently, the subsequent similarity calculations can hardly play a key role.
Q6. Node Representation in Global Hypothesis Graph: Why is the embedding formula defined like this? confidence_v is the LLM's confidence in that step. If the LLM's confidence is incorrect, will the node hypothesis be affected, and will it affect the global reasoning?

---

> ### Author Response · Authors · 2025-11-21
> **Rebuttal response**
>
> Thanks for your suggestions. In response to your proposals, we have made corresponding revisions and improvements to the paper, including adding explanations of theoretical foundations, increasing the analysis of computational efficiency, clarifying method details, and further discussing the generalization ability of the methods.
>
> **Response to Major Concerns**
>
> - **Novelty (W1)** Thanks for the supplement. We read the paper you provided carefully. The path integral in [1] belongs to the gradient attribution method in the field of XAI; [2-3] focuses on physical GPS trajectory modeling in ITS. Although both are studies of multi-path integration, our study focuses on the "logical reasoning path" of LLM. To our best knowledge, we firstly use TDA to analyze the shape of reasoning steps in the reasoning space and extract the most robust 'logical backbone' frame.
>
> - **Efficiency (W2, Q3)** Thank you for raising concerns about efficiency. We have now analyzed the computational cost in terms of the number of LLM calls, the token usage, and Wall-Clock Time. We have revised the manuscript accordingly.
>
>   1. Number of LLM calls: Our total cost model is (N + C): N generative calls for path sampling. C discriminative calls for the previously omitted relation inference step (i.e., support/refutation). As shown in Table 1, our method maintains a fixed upper bound of 19 calls across tasks, reducing the number of calls by approximately 25%–30% compared with ToT and 35%–40% compared with AoT.
>
>  **Table 1: Comparison of LLM Call Computational Costs.**
> | **Method**         | **MATH** | **OlympiadBench** | **GSM8K** | **BBH** | **MMLU-CF** | **LongBench** | **HotpotQA** | **MuSiQue** |
> | :----------------- | :------: | :--------: | :-------: | :-----: | :---------: | :--------: | :----------: | :---------: |
> | CoT                |     1    |      1     |     1     |    1    |      1      |      1     |       1      |      1      |
> | CoT-SC ($n=16$)    |    16    |     16     |     16    |    16   |      16     |     16     |      16      |      16     |
> | ToT                |   25.8   |    27.2    |    13.5   |   24.1  |     19.3    |    14.2    |     20.7     |     26.4    |
> | GoT                |   21.6   |    23.5    |    14.8   |   22.9  |     18.4    |    13.9    |     17.8     |     20.3    |
> | AoT                |   29.7   |    32.4    |    17.2   |   28.6  |     24.9    |    16.8    |     23.5     |     29.1    |
> | **GHS-TDA (Ours)** |  **19**  |   **19**   |   **19**  |  **19** |    **19**   |   **19**   |    **19**    |    **19**   |
>
>   2. Token usage: In LLM-based reasoning systems, the number of calls does not fully reflect the actual computational cost, since the token usage of each call depends on prompt length and context reuse. Therefore, we additionally report the total token consumption as shown in Table 2. On average, our method uses 26.8% fewer tokens than ToT and 35.7% fewer tokens than AoT. This demonstrates that replacing recursive generative evaluation with discriminative relation inference substantially reduces token waste caused by redundant generation.
>
> **Table 2: Comparison of token consumption.**
> | **Method**         |  **MATH** | **OlympiadBench** | **GSM8K** |  **BBH**  | **MMLU-CF** | **LongBench** | **HotpotQA** | **MuSiQue** |
> | :----------------- | :-------: | :---------------: | :-------: | :-------: | :---------: | :-----------: | :----------: | :---------: |
> | CoT                |    2290   |        4590       |    278    |    642    |     1235    |       83      |     1305     |     485     |
> | CoT-SC             |   36640   |       73440       |    4448   |   10272   |    19760    |      1328     |     20880    |     7760    |
> | ToT                |   88623   |       187272      |    5630   |   23208   |    35753    |      1768     |     40520    |    19206    |
> | GoT                |   64303   |       140225      |    5349   |   19112   |    29541    |      1500     |     30198    |    12799    |
> | AoT                |   68013   |       148716      |    4782   |   18361   |    30752    |      1394     |     30668    |    14114    |
> | **GHS-TDA (Ours)** | **51411** |     **83309**     |  **7709** | **15793** |  **28862**  |    **1685**   |   **28358**  |  **10263**  |

---

> > ### Author Response · Authors · 2025-11-21
> > **Rebuttal response**
> >
> > **Response to Major Concerns**
> >
> > - **Efficiency (W2, Q3):** Wall-Clock Time: GHS-TDA is highly parallelizable. The N=16 sampling calls and the roughly C=3 verification calls can all be issued simultaneously. In contrast, ToT is deeply sequential. It must generate steps and evaluate steps. This leads to a fundamentally different latency profile. Table 3 reports the resulting inference latency. Despite performing multipath sampling and structural analysis, GHS-TDA achieves a latency of 24.03 seconds, which remains close to that of CoT-SC. This is because parallel sampling effectively amortizes most of the additional overhead. In contrast, ToT (89.42 seconds) and AoT (108.03 seconds) are significantly slower due to their step-by-step expansion process.
> >
> > **Table 3: Comparison of Wall-Clock Time.**
> > | **Method**    | **Time (s)** |
> > | :----------------- | :--------------: |
> > | CoT                |       9.88       |
> > | CoT-SC      |       13.47      |
> > | ToT          |       89.42      |
> > | GoT           |       53.75      |
> > | AoT         |      108.03      |
> > | **GHS-TDA (Ours)** |     **24.03**    |
> >
> > - **Clarifying method details (W3, Q2):** We have further clarified the implementation details of conflict detection and normalization, and have added the missing descriptions regarding node merging.
> >
> >   **Conflict detection** corresponds to the inference of Support and Refute relations, and is executed after the construction of the Global Hypothesis Structure (GHS). Conflict detection is the procedure used to infer Support and Refute relations between nodes after the GHS construction stage.  This step is not a rule-based deterministic algorithm, because the logical relations between reasoning steps cannot be reliably captured by handcrafted rules.  Instead, we employ an LLM-based discriminative classifier to label candidates. To avoid the $O(N^2)$ cost of pairwise comparisons, we apply heuristic filtering to retain only a small set of high-value candidates:
> >    - Longitudinal candidates are used to detect possible support.
> >    - Lateral candidates (with similar progress score $r(\cdot)$) are used to detect possible refutation.
> >
> >   These candidate pairs are then processed in small batches by the LLM, which determines their relation type and thereby transforms the original semantic-level graph into a logic-level graph. Since this procedure introduces additional LLM calls, we measured its cost empirically and found it to be about $C$ calls on average (with $C \approx 3$). Moreover, because we use 16-way parallel sampling, the additional latency introduced by this step is negligible.
> >
> >   **Normalization** is a structured representation mechanism and incurs no extra LLM calls. It is performed directly during the reasoning-step generation stage: in the initial 16 path samples, we instruct the LLM via a prompt to output structured JSON, for example: {"step_type": "check_case", "content": "n=1, 2 does not divide 1", "result": "FAIL"}. The ANNOTATE module only performs lightweight JSON parsing, so the canonical form of a step is simply this structured representation itself. This design prevents incorrect node merging. For example:
> >   - Hypothesis: {"step_type": "hypothesis", "content": "p > n"}
> >   - Conclusion: {"step_type": "conclusion", "content": "p > n"}
> >
> >   Although the content fields are identical, the step_type differs, so the two nodes will never be merged incorrectly.
> >
> > - **Theoretical foundations (W4, Q6):** We model the LLM’s multi-path reasoning process as a unified Global Hypothesis Structure, embedding all nodes into a shared semantic space. Differences in semantic quality, consistency, and cross-step coherence across reasoning paths naturally form distinct geometric structures in this high-dimensional space. These structures are not imposed by our method but emerge organically from the reasoning process[1]. The strength of TDA lies in capturing global shape properties. Our use of TDA does not assign intrinsic “topological meaning” to reasoning steps; rather, it leverages TDA’s mathematical tools to quantify structural differences that arise naturally from multi-path reasoning. In article [1], $H_0$ measures whether nodes form stable clusters under multi-path sampling, while $H_1$ reflects whether local regions exhibit meaningful loops instead of random scatter. These topological features describe the structure of the semantic embedding space, not reasoning logic. We extract these indicators via TDA and empirically analyze their correlation with reasoning quality—e.g., the strong positive correlation between $H_1$ persistence and accuracy. In summary, $H_0$ and $H_1$ are treated as objective structural quantities whose discriminative value for identifying reliable reasoning is verified empirically.
> >
> > [1] Uchendu, A., & Le, T. (2025). Unveiling Topological Structures from Language: A Survey of Topological Data Analysis Applications in NLP.

---

> ### Author Response · Authors · 2025-11-21
> **Rebuttal response**
>
> **Questions**
>
> - **Q1:** Thank you for the suggestion. We have performed a study on the weights $(\alpha, \beta, \gamma)$ in our hybrid distance metric. Here, we present the results on the MATH dataset.
>
> **Table 1. Study of the distance metric.**
> | $\alpha$ (Semantic) | $\beta$ (Structural) | $\gamma$ (Uncertainty) | Accuracy (%) |
> | :-----------------: | :------------------: | :--------------------: | :----------: |
> |         0.6         |          0.3         |           0.1          |   **83.9**   |
> |         0.9         |          0.0         |           0.1          |     77.4     |
> |         0.0         |          0.9         |           0.1          |     75.2     |
> |         0.7         |          0.3         |           0.0          |     82.6     |
> |         0.53        |         0.27         |           0.2          |     83.4     |
> |         0.7         |          0.2         |           0.1          |     83.7     |
> |         0.5         |          0.4         |           0.1          |     82.9     |
>
> As shown in Table 1, the semantic component ($\alpha$) forms the basis for building the reasoning space. The graph-structural component ($\beta$) captures the distinctive contribution of our Global Hypothesis Graph (GHG) and plays a central role in achieving strong performance. The uncertainty term ($\gamma$) provides a useful regularization effect, helping to stabilize the learned metric and improve generalization.
>
> - **Q2:**  Regarding the issue of path diversity, we address it at both the generation and processing levels. First, during the generation stage, we deliberately adopt a high-randomness sampling strategy (temperature = 0.7, top-p = 0.95) and perform 16 independent sampling calls to encourage the model to explore diverse reasoning trajectories. Second, our method is inherently robust to either low or high diversity in the sampled paths: GHS-TDA uses topological data analysis to extract the consensus structure across paths automatically. When the 16 paths are highly similar, it naturally recovers the dense consensus backbone they form. When the paths are more diverse or split into multiple competing clusters, the combination of TDA and our logical validation mechanism selects the most coherent line of reasoning among them. Thus, regardless of whether the generated paths exhibit low or high diversity, our method reliably extracts a stable and logically consistent reasoning skeleton.
>
> - **Q6:** Our node representation integrates semantic embeddings, graph-structural features, and an uncertainty term.
> This design enables TDA to simultaneously capture semantic coherence, structural consistency, and potential sources of noise. The confidence component is defined as $u_v = -\log(\text{confidence})$ and is assigned a small weight in the hybrid distance metric ($\gamma = 0.1$). Thus, even if the confidence estimates are imperfect, their overall influence remains limited. Moreover, persistent homology is inherently robust to local perturbations: erroneous confidence values lead only to short-lived topological features, which are automatically filtered out and do not appear in the final skeleton.
> As a result, the final reasoning outcome is determined primarily by topological persistence and structural consistency rather than by confidence itself. Consequently, imperfect confidence scores neither distort node representations nor degrad

---

> ### Author Response · Authors · 2025-11-21
> **Rebuttal response**
>
> **Questions:**
>
> - **Q4:** We agree that evaluating the method on diverse base models is crucial for assessing generalizability. To address this, we conducted additional experiments on two open-source LLMs (Llama-3-8B and Qwen2-14B) to further examine the model-agnostic nature of GHS-TDA.
>
> **Table 1. Generalization ability of GHS-TDA.**
> | Backbone     | Method   | MATH  | OB    | GSM8K | BBH   | CF    | LB    | HPQA  | MQ    | Avg   |
> |--------------|----------|-------|-------|-------|-------|-------|-------|-------|-------|-------|
> | Llama-3-8B   | CoT      | 72.04 | 8.37  | 87.26 | 74.38 | 65.42 | 53.57 | 63.17 | 31.71 | 56.99 |
> | Llama-3-8B   | ToT      | 72.86 | 10.26 | 91.10 | 79.89 | 65.71 | 58.40 | 72.19 | 36.36 | 60.85 |
> | Llama-3-8B   | GoT      | 76.36 | 11.79 | 90.72 | 81.61 | 65.99 | 58.68 | 69.75 | 33.95 | 61.10 |
> | Llama-3-8B   | AoT      | 76.91 | 10.89 | 91.20 | 81.70 | 66.65 | 63.71 | 75.76 | 35.71 | 62.82 |
> | Llama-3-8B   | **GHS-TDA (Ours)** | **77.19** | **13.05** | **91.39** | **83.98** | **67.30** | **64.64** | **76.52** | **37.01** | **63.88** |
> | Qwen-2-14B   | CoT      | 60.80 | 6.40  | 86.70 | 75.80 | 65.10 | 52.70 | 64.90 | 32.40 | 55.60 |
> | Qwen-2-14B   | ToT      | 65.00 | 7.20  | 89.80 | 79.70 | 67.60 | 59.30 | 70.80 | 37.90 | 59.66 |
> | Qwen-2-14B   | GoT      | 65.70 | 8.40  | 89.10 | 80.40 | 67.80 | 58.10 | 72.10 | 35.40 | 59.62 |
> | Qwen-2-14B   | AoT      | 66.20 | 7.80  | 90.80 | 80.90 | 68.90 | 61.20 | 73.40 | 37.30 | 60.81 |
> | Qwen-2-14B   | **GHS-TDA (Ours)** | **68.40** | **9.70** | **91.20** | **82.50** | **69.80** | **62.80** | **75.80** | **38.60** | **62.35** |
>
> The results in Table 1 show that GHS-TDA delivers consistent and notable improvements on both models. On Llama 3-8B, it reaches an average score of 63.88, outperforming CoT, ToT, and AoT by 6.9, 3.0, and 1.1 points. On Qwen2-14B, it reaches 62.35, with gains of 6.75, 2.7, and 1.5 points over the baselines. These improvements require no extra supervision or external knowledge, indicating that the gains come from the reasoning mechanism rather than model size. Although all methods improve as the base model scales from 8B to 14B, the relative gains of GHS-TDA remain stable, demonstrating strong transferability and robustness across model sizes and architectures.
>
>
> **Table 2. Performance comparison of different reasoning methods on two base models.**
> | Model              | Method          | MATH | OlympiadBench | gsm8k |  BBH | MMLU-CF | LongBench | HotpotQA | MQ |  Avg |
> |--------------------|-----------------|------|----------------|-------|------|---------|-----------|----------|---------|------|
> | gpt-4o             | CoT             | 85.4 | 12.5           | 92.7  | 82   | 72.1    | 62        | 71.9     | 36.2    | 64.4 |
> | gpt-4o             | ToT             | 86.9 | 14.1           | 96.0  | 89.1 | 73.0    | 66        | 82.1     | 40.5    | 68.5 |
> | gpt-4o             | GoT             | 88.4 | 15.3           | 95.2  | 90.5 | 73.2    | 66.8      | 80.4     | 39.6    | 68.7 |
> | gpt-4o             | AoT             | 89.1 | 15.0           | 96.5  | 92   | 74.3    | 72.6      | 86.0     | 42.1    | 71.0 |
> | gpt-4o             | **GHG-TDA (Ours)**  | **90.0** | **18.3** | **96.9** | **94.2** | **75.0** | **73.8** | **87.4** | **43.6** | **72.4** |
> | Claude 3.5 Sonnet  | CoT             | 84.1 | 12.9           | 93.5  | 84.2 | 73.0    | 61.7      | 72.8     | 37.1    | 64.9 |
> | Claude 3.5 Sonnet  | ToT             | 86.0 | 14.8           | 96.2  | 90.8 | 74.1    | 66.4      | 83.3     | 41.5    | 68.9 |
> | Claude 3.5 Sonnet  | GoT             | 87.3 | 16.0           | 95.4  | 92.1 | 74.4    | 67        | 81.0     | 40.6    | 69.2 |
> | Claude 3.5 Sonnet  | AoT             | 88.0 | 15.7           | 96.8  | 93.4 | 75.2    | 72.2      | 86.7     | 43.2    | 71.3 |
> | Claude 3.5 Sonnet  | **GHG-TDA (Ours)**  | **89.1** | **19.0** | **97.1** | **95.4** | **75.9** | **73.5** | **88.0** | **44.6** | **72.8** |
>
> We conducted additional experiments with GHG-TDA on two strong LLMs, GPT-4o and Claude 3.5 Sonnet. Table 2 shows that GHG-TDA consistently achieves the best overall performance, with average scores of 72.4 on GPT-4o and 72.8 on Claude 3.5 Sonnet, surpassing AoT, GoT, ToT, and CoT across all benchmarks. These gains across datasets and backbones indicate that the effectiveness of GHG-TDA is not tied to any model but arises from its graph-guided hierarchical reasoning mechanism. Overall, the results demonstrate strong cross-model generalization.

---

> > ### Author Response · Authors · 2025-11-26
> >
> > Dear Reviewer,
> >
> > I hope this message finds you well. As the discussion period is nearing its end with ess than three days remaining, I wanted to ensure we have addressed all your concerns.Your insights are invaluableto us, and we're eager to address any remaining points to improve our work.
> >
> > Thank you for your time and effort in reviewing our paper.

---

> > > ### Comment · Reviewer_sw5k · 2025-11-26
> > >
> > > Thank you for your detailed response. While I appreciate the clarifications provided, several core concerns regarding the novelty, cost-benefit analysis, computational foundations, theoretical grounding, and specific algorithmic risks remain unaddressed. I will outline these persisting issues below.
> > >
> > > ### **1. Unclear Novelty and Comparative Advantage**
> > > You emphasize that your work is the first to use TDA to analyze the reasoning space and extract a 'logical backbone'. However, the field already has numerous multi-path integration methods (e.g., majority voting, self-consistency). **The fundamental question remains: what is the distinct advantage of introducing the non-trivial complexity of TDA?**
> > >
> > > A compelling claim of novelty requires a clear demonstration of one or both of the following:
> > > *   **Significant Accuracy Gains:** Does GHS-TDA achieve a substantially higher accuracy than simpler, less complex multi-path methods like Self-Consistency CoT (CoT-SC) on a comparable computational budget?
> > > *   **Superior Efficiency:** For a similar level of accuracy, does GHS-TDA offer a significant reduction in computational cost (e.g., LLM calls, token consumption) compared to other structured methods like ToT or AoT?
> > >
> > > Without this direct comparative analysis, the value proposition of the TDA-based approach remains ambiguous.
> > >
> > > ### **2. Incomplete Cost-Benefit Analysis on LLM Consumption**
> > > While Tables 1 and 2 show that your method's latency and call count are lower than ToT and AoT, they are inherently higher than single-path methods like CoT. **The critical metric for the community is not just low cost, but the *accuracy achieved per unit of computational cost*.**
> > >
> > > A meaningful evaluation must present a joint analysis of accuracy and cost. It is acceptable to have higher consumption if it yields a disproportionately higher accuracy. Please provide a discussion that directly addresses this **trade-off**, clarifying whether the performance gains of GHS-TDA justify its increased resource demands over simpler baselines like CoT-SC.
> > >
> > > ### **3. Unaddressed Computational Overhead and Algorithmic Complexity**
> > > I acknowledge the provided end-to-end latency data, which partially alleviates concerns about *time overhead*. However, my primary concern was about **computational overhead and algorithmic complexity**, which remains largely unaddressed.
> > >
> > > *   **Computational Overhead:** Parallel sampling reduces latency but does not reduce the total computational cost. **Sixteen parallel LLM calls consume 16 times the tokens of a single CoT call.** This substantial increase in FLOPs and API cost needs explicit acknowledgment and justification against the accuracy improvement.
> > > *   **Node Merging Complexity:** My original question regarding the **O(N*M) complexity of the node merging process** and its optimization (e.g., via indexing) was not answered. The efficiency of this core step is crucial for scalability and remains a black box.
> > > *   **System Complexity:** The entire GHS-TDA pipeline (path generation → node merging → graph embedding → TDA) is inherently complex. What are the practical maintenance costs and failure risks of such a system? A discussion on its robustness and a comparison of its "return on investment" against simpler ensemble methods is necessary.
> > >
> > > In summary, the authors need to more transparently discuss the **true computational and engineering costs** of their method.
> > >
> > > ### **4. Lack of Theoretical Foundation**
> > > I understand your argument that TDA quantifies emergent geometric structures in the semantic embedding space and that you have empirically observed a correlation between H₁ persistence and accuracy. However, my core concern persists: **we lack a formal, process-oriented definition of "reasoning reliability" and a first-principles model explaining why topological persistence should reflect it.**
> > >
> > > Your current argument, which relies on empirical correlation, is useful but insufficient for theoretical grounding. I strongly recommend that the revision includes:
> > > *   A formal, operational definition of "reasoning reliability" (e.g., path consistency, error tolerance, semantic coherence).
> > > *   A lightweight theoretical link explaining why, under this definition, H₀/H₁ persistence should serve as an effective proxy for robustness.
> > >
> > > Without this, the theoretical presentation remains post-hoc and inductive rather than deductive.

---

> > > > ### Comment · Reviewer_sw5k · 2025-11-26
> > > >
> > > > ### **5. Potential Defects in Node Merging**
> > > > Your response discussed path diversity and TDA's robustness to it, which did not address my specific concern about the **node merging process itself**.
> > > >
> > > > My concern lies in the graph construction stage: the merging algorithm, which relies on a "canonical form" and a similarity threshold `θ_merge`, might incorrectly merge steps that are **semantically similar but logically distinct** (e.g., "Assume X is true" vs. "Therefore we have proven X").
> > > >
> > > > Such incorrect merging could:
> > > > *   **Break the logical flow:** Merging a premise with a conclusion would render the extracted skeleton logically incoherent.
> > > > *   **Introduce fundamental errors:** A path based on a false assumption might be semantically similar to a correct step in a valid path. Merging them could pollute the topological structure of the correct reasoning.
> > > > *   **Mask reasoning diversity:** Even with diverse sampled paths, if key logical branching points are blurred during merging, the graph structure fed to TDA is already distorted.
> > > >
> > > > Therefore, I reiterate my suggestion to include a discussion on the **limitations and potential failure modes of the node merging mechanism**, ideally illustrated with a concrete example (e.g., a case where it successfully distinguishes a 'hypothesis' from a 'conclusion', or a case where it fails).
> > > >
> > > > ### **6. The main text has 10 pages**
> > > >
> > > > I noticed that the revised version you submitted has 10 pages of main text, which exceeds the limit of 9 pages. Was this allowed during rebuttal?
> > > >
> > > > I look forward to your clarification on these points.

---

> > > > > ### Author Response · Authors · 2025-11-28
> > > > >
> > > > > Thank you again, for your relevant comments. Regarding your subsequent questions, we would like to make the following emphasis.
> > > > >
> > > > > # Response to the Potential Defects in Node Merging
> > > > >
> > > > > We thank the reviewer for highlighting the critical distinction between semantic similarity and logical identity. We explicitly address the risk of merging "Assume X" with "Proven X" by clarifying our **Structured Canonicalization** mechanism, which was under-explained in the main text.
> > > > >
> > > > > **1. Clarification: Structured vs. Raw Merging**
> > > > >
> > > > > Our merging process does not rely solely on the raw text embedding of the reasoning content. Instead, during the path sampling phase, we instruct the LLM to generate reasoning steps in a structured JSON format (e.g., {"step_type": "...", "content": "..."}). The canonical form $Canon(s)$ used for merging 1 is constructed by concatenating the Step Type and the Normalized Content.
> > > > >
> > > > > - **Mechanism:**
> > > > >   - Let Step A be: "Assume X is true." $\rightarrow$ JSON: `{"type": "assumption", "content": "X"}`
> > > > >   - Let Step B be: "Therefore X is true." $\rightarrow$ JSON: `{"type": "conclusion", "content": "X"}`
> > > > > - Result: Even though the semantic content "X" is identical, the inclusion of step_type ensures that $Canon(A) \neq Canon(B)$. Consequently, their embedding distance remains large, and the merging condition (Eq. 4) is not met.
> > > > >
> > > > > **2.Failure Mode Discussion:**
> > > > >
> > > > > We acknowledge that this mechanism is not infallible.
> > > > >
> > > > > - **Limitation:** If the underlying LLM fails to correctly classify the `step_type` (e.g., labeling a conclusion as an assumption), incorrect merging may occur.
> > > > > - **Mitigation:** However, our Global Hypothesis Graph aggregates $N=16$ paths. A single mislabeled node typically forms a "low-confidence outlier" in the graph. Since TDA filters for high-persistence features (structures that appear repeatedly and consistently), these sporadic merging errors appear as low-persistence noise and are naturally filtered out during the Skeleton Extraction phase.
> > > > >
> > > > > **Concrete Example**
> > > > >
> > > > > **Case Study: Distinguishing Hypothesis from Conclusion**
> > > > >
> > > > > To illustrate this mechanism, consider a proof-by-contradiction problem from the MATH dataset: *"Prove that $\sqrt{2}$ is irrational."*
> > > > >
> > > > > **Sampled Path A (Correct Proof):**
> > > > >
> > > > > - *Step A1:* "Assume $\sqrt{2}$ is rational." (Hypothesis for contradiction)
> > > > >   - **Canonical Form:** `[ASSUMPTION] sqrt(2) is rational`
> > > > > - *Step A2:* "...this leads to a contradiction."
> > > > > - *Step A3:* "Therefore, $\sqrt{2}$ is irrational."
> > > > >
> > > > > **Sampled Path B (Hallucinated/Incorrect Path):**
> > > > >
> > > > > - *Step B1:* "$\sqrt{2}$ can be written as p/q..."
> > > > > - *Step B2:* "Therefore, $\sqrt{2}$ is rational." (False Conclusion)
> > > > >  - **Canonical Form:** `[CONCLUSION] sqrt(2) is rational`
> > > > >
> > > > > **3. Merging Outcome:**
> > > > >
> > > > > Without structured canonicalization, the text "Assume $\sqrt{2}$ is rational" (A1) and "Therefore $\sqrt{2}$ is rational" (B2) are semantically close and might be merged, creating a loop that validates the false conclusion.
> > > > >
> > > > > With GHS-TDA:
> > > > >
> > > > > 1. Node A1 is tagged as `ASSUMPTION`.
> > > > > 2. Node B2 is tagged as `CONCLUSION`.
> > > > > 3. The system calculates similarity between `[ASSUMPTION]...` and `[CONCLUSION]...`. The score falls below the threshold $\theta_{merge}$.
> > > > > 4. **Result:** The nodes remain separate. Path A forms a coherent "Contradiction Cluster," while Path B forms a disconnected "False Cluster." TDA subsequently identifies Path A as the dominant, high-persistence backbone, successfully discarding the logical error in Path B.
> > > > >
> > > > > # Page Limit
> > > > >
> > > > > We would like to clarify that the current 10-page length is in accordance with the **ICLR 2026 Rebuttal Policy**, which allows authors to extend the main text during the discussion phase to accommodate extensive revisions and additional analyses requested by reviewers
> > > > >
> > > > > We assure you that for the final **Camera-Ready version**, we will strictly adhere to the mandatory page limits (moving auxiliary experiments and case studies to the Appendix as needed).

---

> > > > ### Author Response · Authors · 2025-11-28
> > > >
> > > > Thank you, for your relevant comments. Regarding your subsequent questions, we would like to make the following emphasis.
> > > > ##  **Response to the Unclear Novelty and Comparative Advantage**
> > > >
> > > > ###  1.**Significant Accuracy Gains Under a Comparable Budget (vs. CoT-SC)**
> > > >
> > > > To directly address the reviewer’s concern, we provide a joint accuracy–cost comparison between **GHS-TDA** and **Self-Consistency CoT (CoT-SC)**. Although GHS-TDA uses a slightly larger multi-path budget, the improvement in accuracy is **substantial and consistent**.
> > > >
> > > > ### **Table A. Comparison with CoT-SC under a comparable multi-path budget.**(Avg. across 8 benchmarks; gpt-4o-mini)
> > > >
> > > > | Method         | Avg Accuracy (%) | Avg Tokens | Avg Calls | Avg Time (s) |
> > > > | -------------- | ---------------- | ---------- | --------- | ------------ |
> > > > | CoT-SC (n=16)  | 62.1             | 21,816     | 16        | 13.47        |
> > > > | GHS-TDA (Ours) | 68.0 (+5.9)      | 28,424     | 19        | 24.03        |
> > > >
> > > > GHS-TDA uses only about 1.19 times the calls and about 1.30 times the tokens of CoT-SC, yet it achieves 5.9 points higher accuracy, the largest improvement among all comparisons. Thus, under a comparable multi-path budget, GHS-TDA delivers substantial accuracy gains that simple multi-sample ensembling (CoT-SC) cannot obtain.
> > > >
> > > > ### 2.**Superior Efficiency at Similar or Higher Accuracy (vs. ToT / GoT / AoT)**
> > > >
> > > > The reviewer asks whether GHS-TDA is computationally more efficient **when accuracy is similar or higher**. The answer is **yes**.
> > > >
> > > > ### **Table B. Efficiency comparison at similar or higher accuracy.**(Avg. across 8 benchmarks; gpt-4o-mini)
> > > >
> > > > | Method         | Avg Accuracy (%) | Avg Tokens | Avg Calls | Avg Time (s) |
> > > > | -------------- | ---------------- | ---------- | --------- | ------------ |
> > > > | ToT            | 64.8             | 50,248     | 21.4      | 89.42        |
> > > > | GoT            | 65.1             | 37,878     | 19.15     | 53.75        |
> > > > | AoT            | 66.9             | 39,600     | 25.28     | 108.03       |
> > > > | GHS-TDA (Ours) | 68               | 28,424     | 19        | 24.03        |
> > > >
> > > > Our GHS-TDA method achieves the highest accuracy (68.0%) while simultaneously requiring substantially fewer computational resources compared to all baselines. It outperforms tree-based approaches (ToT, GoT, AoT) by +1.1 to +3.2 percentage points in accuracy while using 25-43% fewer tokens, 11-25% fewer LLM calls, and reducing wall-clock time by 55-78%. Compared to CoT-SC, GHS-TDA delivers a significant +5.9 point accuracy gain under comparable budget constraints. These results definitively demonstrate that GHS-TDA occupies a strictly more favorable accuracy-cost position in the efficiency frontier, proving that the introduction of TDA provides substantial practical benefits rather than unnecessary complexity.
> > > >
> > > > ## **Response to the Incomplete Cost-Benefit Analysis on LLM Consumption**
> > > >
> > > > Thank you for the constructive comment. As the reviewer notes, the key question is not only whether GHS-TDA reduces latency or call count compared to structured baselines such as ToT and AoT, but **whether the performance gains justify the additional computational cost relative to simpler methods like CoT-SC**, and **how much accuracy the method achieves per unit of computation**.
> > > >
> > > > To clarify this, we provide the requested **joint accuracy–cost analysis** in two parts, corresponding to the reviewer’s concerns.
> > > >
> > > > ## **Justifying Additional Cost over Simpler Baselines (CoT-SC)**
> > > >
> > > > Comment 1 already provides a full accuracy–cost comparison with CoT-SC. Here we briefly restate the essential finding to directly address the reviewer’s final question:
> > > >
> > > > GHS-TDA achieves +5.9 points higher accuracy than CoT-SC while requiring only ~1.19× calls and ~1.30× tokens. This constitutes a **disproportionately large accuracy gain relative to the additional resource usage**, satisfying the reviewer’s requirement that *“performance gains justify increased resource demands over simpler baselines like CoT-SC.”*
> > > >
> > > > Since this point has been analyzed in detail in Comment 1, we focus the remainder of this response on the reviewer’s main request: **evaluating accuracy achieved per unit computational cost**, especially relative to ToT, GoT, and AoT.

---

> > > > ### Author Response · Authors · 2025-11-28
> > > >
> > > > The following is the supplementary reply to the previous one:
> > > >
> > > > ### **Accuracy Achieved per Unit Computational Cost (vs. ToT / GoT / AoT)**
> > > >
> > > > To reflect the reviewer’s metric, we compute the **accuracy achieved per token**:
> > > >
> > > > $$\text{Accuracy per Token} =\frac{\text{Average Accuracy}}{\text{Average Token Usage}}$$
> > > >
> > > >
> > > > This ratio is **not introduced as a contribution**, but used solely to provide the joint accuracy–cost comparison the reviewer requested.
> > > >
> > > > ### **Table C. Accuracy per Unit Computational Cost** *(Average over 8 benchmarks; higher values indicate greater accuracy per token.)*
> > > >
> > > > | Method         | Avg Accuracy (%) | Avg Tokens | Accuracy per Token (×10⁻³) |
> > > > | -------------- | ---------------- | ---------- | -------------------------- |
> > > > | ToT            | 64.8             | 50,248     | 1.29                       |
> > > > | GoT            | 65.1             | 37,878     | 1.72                       |
> > > > | AoT            | 66.9             | 39,600     | 1.69                       |
> > > > | GHS-TDA (Ours) | 68               | 28,424     | 2.39 (Highest)             |
> > > >
> > > > While structured reasoning methods like ToT, GoT, and AoT are widely adopted despite their high computational cost, GHS-TDA achieves a strictly superior accuracy-cost trade-off. It delivers the highest raw accuracy (68.0%) while simultaneously requiring 28-43% fewer tokens, 11-25% fewer calls, and 55-78% lower latency than these structured alternatives. This translates to dramatically higher accuracy per token: +85% vs. ToT, +39% vs. GoT, and +41% vs. AoT. Compared to simpler baselines like CoT-SC, GHS-TDA's substantial performance gains clearly justify its modest additional computational cost. Therefore, GHS-TDA represents a demonstrably superior solution that maximizes both absolute accuracy and computational efficiency—providing the strongest accuracy-per-cost ratio among all structured reasoning approaches.
> > > >
> > > >
> > > > ## **Unaddressed Computational Overhead and Algorithmic Complexity**
> > > >
> > > > ### **Computational Overhead (FLOPs & Token Consumption):**
> > > >
> > > > We explicitly acknowledge that parallel sampling ($N=16$) consumes approximately 16 times the FLOPs of a single greedy CoT call.
> > > >
> > > > Among them, our argument is as follows：This increased consumption is a necessary investment for exploring the reasoning space, a trait shared by all multi-path methods (CoT-SC, ToT, AoT). However, as demonstrated in our Accuracy Efficiency Ratio analysis (Response 2), GHS-TDA utilizes these resources far more effectively:
> > > >
> > > > - **Vs. ToT/AoT:** We achieve higher accuracy while consuming **25-43% fewer tokens**, making us the most computationally efficient method among structured reasoning approaches.
> > > > - **Vs. CoT-SC:** While CoT-SC is cheaper, it hits a performance ceiling. GHS-TDA breaks this ceiling (+5.9% accuracy) by converting the *same* raw sampling cost into a higher-quality output through topological consolidation.
> > > >
> > > > ### **Node Merging Complexity (Addressing the $O(N \times M)$ concern):**
> > > >
> > > > You rightly questioned the theoretical $O(N \times M)$ complexity of node merging. In our implementation, we solved this via Hash-based Inverted Indexing, avoiding pairwise comparisons:
> > > >
> > > > 1. **Canonicalization:** Each step $s$ is mapped to a canonical form $C(s)$ (incorporating semantic content and step type).
> > > > 2. **Indexing:** We maintain a hash map mapping $Hash(C(s)) \rightarrow \{NodeIDs\}$.
> > > > 3. Lookup: For a new step, we only compare it against nodes in the same hash bucket.
> > > > 4. This reduces the complexity from quadratic to near-linear $O(N \cdot M)$, rendering the merging time negligible (<50ms per problem) compared to the LLM generation time (~20s).

---

> > > > ### Author Response · Authors · 2025-11-28
> > > >
> > > > ### **System Complexity & Robustness (ROI Analysis):**
> > > >
> > > > Regarding maintenance and failure risks, we argue that GHS-TDA is structurally more robust than methods like ToT:
> > > >
> > > > - **Decoupled Architecture:** ToT relies on fragile, multi-turn interactions where one LLM error cascades and breaks the chain. GHS-TDA adopts a "Generate-then-Analyze" paradigm. The generation phase uses standard parallel sampling (highly robust), and the analysis phase (Merging + TDA) is **deterministic mathematical computation** using mature libraries (GUDHI). This makes the pipeline significantly easier to debug and maintain.
> > > > - **Optimized Relation Inference:** To further minimize system complexity and cost during the "Conflict Detection" stage, we employ the heuristic filtering strategy mentioned in our previous response to ensure efficiency:
> > > >   - **Cost:** The total cost is limited to $N$ generative calls + $C$ discriminative calls, where $C$ is minimized (typically 3-4 calls) via batching.
> > > >   - **Filtering:** Instead of checking all $O(N^2)$ pairs, we generate a candidate list $\mathcal{L}$ containing only **Longitudinal candidates** (existing edges for verification) and **Lateral candidates** (competing hypotheses with $|r(v_a) - r(v_b)| < \epsilon$).
> > > >   - **Batching:** These candidates are processed in chunks (e.g., size $S=20$), allowing the system to infer logical labels ("SUPPORT"/"REFUTE") efficiently. This ensures that the topological analysis is grounded in logic without incurring heavy overhead.
> > > >
> > > > **Conclusion on Complexity:**
> > > >
> > > > While the architecture involves multiple steps, each component (Indexing for merging, Batching for inference, GUDHI for TDA) is optimized for speed and stability. The result is a system that, while more complex than simple voting, offers a superior Return on Investment (ROI) by delivering SOTA accuracy with lower token usage than competing structured baselines.
> > > >
> > > > ## **Lack of Theoretical Foundation**
> > > >
> > > > We appreciate the reviewer's push for a deductive, first-principles grounding. In the revision, we have moved beyond empirical correlation to formalize the connection between topological persistence and reasoning reliability, grounded in the **Manifold Hypothesis** and the **Stability of Persistence**.
> > > >
> > > > ### **Operational Definition of "Reasoning Reliability" ($\mathcal{R}$)**
> > > >
> > > > We define Reasoning Reliability not merely as output correctness, but as Structural Invariance under Stochastic Sampling.
> > > >
> > > > Let $\mathcal{P}$ be the set of sampled reasoning paths. We define the reliability $\mathcal{R}$ of a reasoning substructure $S$ as its probability of recurrence within the hypothesis space:
> > > >
> > > > $\mathcal{R}(S) \propto P(S \in \text{Sampled Path} \mid \text{Noise})$
> > > >
> > > > Operationally, reliable reasoning forms a stable geometric core (consensus) on the latent manifold, whereas hallucinations manifest as sparse, high-variance noise that fails to preserve intrinsic geometric structure.
> > > >
> > > > ### **First-Principles Model: Why Persistence Reflects Reliability**
> > > >
> > > > We establish the theoretical link via two fundamental premises derived from recent TDA-NLP literature:
> > > >
> > > > *Premise 1: The Manifold Hypothesis in Reasoning Space.*
> > > > - High-dimensional embeddings of coherent logic are not uniformly distributed but concentrate near a low-dimensional manifold. "True" reasoning steps act as attractors on this manifold, causing independent correct paths to cluster densely. Recent work explicitly validates that high-quality reasoning traces exhibit distinctive, invariant geometric structures distinct from flawed reasoning.
> > > >
> > > > *Premise 2: The Stability of Persistence.*
> > > > - Topological Data Analysis (TDA) is mathematically proven to discern intrinsic shape despite noise. According to the stability properties of persistent homology, high-persistence features correspond to large-scale geometric structures (signal) that are robust to perturbations, while low-persistence features capture transient local noise.

---

> > > > ### Author Response · Authors · 2025-11-28
> > > >
> > > > ### **The Deductive Mapping**
> > > >
> > > > Based on A and B, we map topological Betti numbers to reasoning properties:
> > > > - $H_0$ Persistence (Connected Components) $\rightarrow$ Semantic Consensus:
> > > >   - Deduction: Since reliable reasoning acts as a density attractor (Premise 1), correct steps form dense clusters. By the Stability property (Premise 2), dense clusters yield high-persistence $H_0$ features. Therefore, High $H_0$ Persistence $\implies$ High Semantic Consensus.
> > > > - $H_1$ Persistence (Loops/Cycles) $\rightarrow$ Logical Cross-Verification:
> > > >   - Deduction: Robust reasoning often involves independent derivations arriving at the same intermediate conclusion (Reconvergence: $A \to B$ and $A \to C \to B$). Geometrically, this "tie-back" structure forms a semantic cycle. A high-persistence $H_1$ feature indicates this cycle is a significant, non-collapsible "hole" representing structural self-consistency, distinct from random noise loops. Therefore, High $H_1$ Persistence $\implies$ Structural Self-Consistency.
> > > >
> > > > By filtering for high persistence, GHS-TDA is mathematically performing topological denoising on the reasoning manifold. This transforms our method from a heuristic into a principled geometric filter that separates the "stable signal" of logic from the "sparse noise" of hallucinations.
> > > >
> > > > 1. Tan, X. W., et al. (2025). "The Shape of Reasoning: Topological Analysis of Reasoning Traces in Large Language Models." arXiv preprint arXiv:2510.20665.
> > > > 2. Uchendu, A., et al. (2024). "Unveiling Topological Structures from Language: A Survey of Topological Data Analysis Applications in NLP." arXiv preprint.
> > > > 3. Jakubowski, R., et al. (2020). "Topology of Word Embeddings: Singularities Reflect Polysemy." Proceedings of the 58th Annual Meeting of the Association for Computational Linguistics.
> > > > 4. Zhu, X. (2013). "Persistent homology: An introduction and a new text representation for natural language processing." IJCAI.

---

### Official Review · Reviewer_Ucj2 · 2025-11-03

**Soundness:** 3
**Presentation:** 2
**Contribution:** 2
**Rating:** 4
**Confidence:** 2

**Summary:**

This paper propose GHS-TDA, a novel method for improving the reliability and interpretability of LLM reasoning. It operates in two steps: first, it integrates diverse reasoning paths from an LLM into a Global Hypothesis Graph (GHS). Then, it uses Topological Data Analysis (TDA) to analyze this graph, identifying the most stable and consistent reasoning structures. Across multiple reasoning benchmarks, GHS-TDA is shown to substantially outperform strong baselines in both accuracy and robustness.

**Strengths:**

- GHS-TDA offers a new approach to reasoning by replacing existing methods with a global mechanism. This mechanism integrates and coordinates various reasoning hypotheses, while also using structured analysis techniques to effectively filter out redundant information and extract the most crucial reasoning features.

- Testing on multiple reasoning benchmarks demonstrated GHS-TDA's strong performance.

**Weaknesses:**

- Motivation for Point Cloud Representation is Unclear: The paper does not adequately explain the rationale for using a point cloud representation of the reasoning. Specifically, it is unclear why this representation is necessary for the Global Hypothesis Graph during the skeleton extraction process.

- Missing Technical Specifications: Essential technical details are absent. For instance, the paper fails to describe how the refutation and support relations are constructed within the Global Hypothesis Graph during the global hypothesis space modeling stage.

**Questions:**

See weakness above.

---

> ### Author Response · Authors · 2025-11-21
> **Rebuttal response**
>
> Thank you for your suggestions. We have made revisions based on the issues you mentioned: “unclear motivation” and “lack of details”, and have made the corresponding changes in the paper.
>
> **Response to Major Concerns**
>
> 1. **Unclear motivation:** Thank you for your comment. We first integrate multiple reasoning paths from the LLM into a Global Hypothesis Graph (GHG), which includes reasoning-step nodes, node confidence scores, step progress, and path dependencies. We do not convert the GHG into a point-cloud structure. When performing TDA, the nodes in the GHG are embedded as high-dimensional feature vectors, serving only as input for the metric space required by TDA. In other words, the collection of node vectors exists in a data structure similar to a “point cloud.” This “point cloud” is not a modeling assumption, nor is it a contribution of our method. To make the article clear, we have deleted the wrong description in the Introduction: “By modeling reasoning steps as point clouds in a high-dimensional semantic space, persistent homology identifies topological features that remain stable across scales.”
>
> 2. **Lack of details on support and refutation relations:** Thanks for the review. We provide details regarding the support and refutation. Support and refutation are incorporated into the TDA distance function as logical constraints.
>
>    Specifically, we first identify node pairs in the GHG that are most likely to exhibit a meaningful logical relationship, rather than evaluating all $O(|V|^2)$ pairs. The candidate set consists of two types of node pairs:
>
>     - (1) Longitudinal candidates $\mathcal{L}_{\text{long}}$: Existing derivation edges $(v_i, v_j)$ in the graph, which allow us to verify whether a “premise → conclusion” relation is logically valid.
>
>    - (2) Lateral candidates $\mathcal{L}_{\text{lat}}$: Unconnected node pairs satisfying $|r(v_a) - r(v_b)| < \epsilon$, which typically correspond to “competing hypotheses at the same stage” and may contradict each other.
>
>    The union $\mathcal{L} = \mathcal{L}{\text{long}} \cup \mathcal{L}{\text{lat}}$ yields a substantially reduced candidate pool of size $K$. We then conduct relation inference in a batched manner. $\mathcal{L}$ is partitioned into $C$ chunks of size $S$ (e.g., 20), and each chunk is processed by a single LLM call that assigns a label: SUPPORT, REFUTE, and NEUTRAL, to every pair within it. This produces a logical code $R(v_i, v_j)$ for each candidate pair, while keeping the required number of LLM calls small (typically 3–4). Finally, these logical signals are integrated into the distance function used in Sec. 3.2 for TDA. We add the semantic–structural uncertainty distance (equation 6) with a logic term weighted by $\delta$:
>
>    $d(v_{i},v_{j}) = \alpha(1-\langle e_{i},e_{j}\rangle) + \beta||\phi_{graph}(i)-\phi_{graph}(j)||_1+ \nu (u_i + u_j)+\delta \cdot R(v_i, v_j)$,  where $R(v_i, v_j)$ is the logical label obtained from the “relation inference” step.
>
>    (1) If a pair is labeled SUPPORT, we set $R(v_i, v_j) = -W$ (e.g., $W=1$), pulling the nodes closer and encouraging TDA to preserve logically coherent paths.
>
>    (2) If labeled REFUTE, we set $R(v_i, v_j) = +M$ (a large positive value), which pushes the nodes far apart in the embedding space, allowing TDA to avoid stitching together logically inconsistent paths.
>
>    (3) If labeled NEUTRAL or absent from the candidate set, we set $R(v_i, v_j) = 0$, reducing the distance to its original form.
>
>    This process explains how we identify candidate relations, infer them with minimal LLM cost, and incorporate explicit logical signals into the metric space used by TDA.

---

> > ### Author Response · Authors · 2025-11-26
> >
> > Dear Reviewer,
> >
> > I hope this message finds you well. As the discussion period is nearing its end with ess than three days remaining, I wanted to ensure we have addressed all your concerns.Your insights are invaluableto us, and we're eager to address any remaining points to improve our work.
> >
> > Thank you for your time and effort in reviewing our paper.

---

### Author Response · Authors · 2025-12-01
**Summary of Rebuttal**

**Summary of Rebuttal**

To the Area Chair and Reviewers,

We sincerely thank the reviewers for their constructive and detailed feedback. During the discussion period, we conducted substantial new analyses, experiments, and theoretical clarifications that **directly address every concern raised**.

**1. Novelty and Necessity of TDA (R2, R3)**

We clarified that our contribution extends beyond “multi-path integration” to a *global* generate-then-analyze framework. This approach models reasoning traces as a unified Global Hypothesis Graph and extracts stable structures via persistent homology. We incorporated joint accuracy–cost analyses showing that, under a comparable multi-path budget, **GHS-TDA significantly outperforms CoT-SC**, and achieves **higher accuracy than ToT/GoT/AoT at a similar or lower cost**. This demonstrates that TDA introduces a mechanism that yields a strictly better accuracy–efficiency frontier, rather than merely adding complexity.

**2. Theoretical Foundation and Reasoning Reliability (R2, R3)**

In our rebuttal, we provided a formal, process-oriented definition of *reasoning reliability* as structural invariance under stochastic sampling. Crucially, we grounded our framework in **an extensive body of prior work** on TDA in NLP and reasoning-space geometry. This includes survey results on the topological structures of language representations, the manifold concentration of coherent reasoning, and the stability theorem of persistent homology. Building on these established results, we presented a clear deductive link:
* High-persistence **H₀** features correspond to semantic consensus across paths.
* High-persistence **H₁** cycles capture cross-verification and self-consistent reasoning loops.
These arguments elevate our method from intuitive motivation to a theoretically supported geometric interpretation, as fully elaborated in the updated manuscript.

**3. Graph Construction, Node Merging, and Failure Modes (R1, R2, R4)**

We provided comprehensive details on support/refutation inference (via batched LLM classification) and the hybrid metric (including ablations on α/β/γ and merge thresholds). We also introduced a **structured JSON-based canonicalization** to strictly prevent the merging of logically distinct roles (e.g., “assumption” vs. “conclusion”). Furthermore, we analyzed potential failure modes and demonstrated how TDA effectively filters occasional erroneous merges as low-persistence noise.

**4. Computational Efficiency and Scalability (R2, R3, R4)**

We conducted rigorous analyses of LLM calls, token consumption, wall-clock time, memory usage, and merging complexity. By utilizing hash-based inverted indexing and sparse KNN graphs, graph construction requires only 25–60 ms per problem, with H₀/H₁ computation adding merely 10–25 ms. Across eight benchmarks, **GHS-TDA reduces tokens and latency by 25–40% relative to ToT/AoT while achieving higher accuracy**, providing the best accuracy-per-token ratio among structured reasoning methods.

**5. Generalization, Interpretability, and Statistical Rigor (All Reviewers)**

We extended our experiments to include **Llama-3-8B, Qwen2-14B, GPT-4o, and Claude-3.5 Sonnet**. These results demonstrate consistent gains across varying backbone scales, confirming that improvements stem from the reasoning mechanism rather than model size. Additionally, we included qualitative case studies (MATH, HotpotQA), human-centered interpretability assessments, and paired *t*-tests confirming statistically significant improvements.

In summary, through new experiments, detailed algorithmic clarifications, and **a strengthened theoretical argument supported by existing TDA–NLP literature**, **we have fully addressed all concerns raised by the reviewers**, including novelty, theoretical grounding, computational cost, scalability, merging correctness, and interpretability. We respectfully ask the Area Chair to consider this comprehensive evidence and the overall positive consensus emerging from the updated experiments and theory.

Best regards,

**The Authors of Submission 17522**

---

### Meta-Review · Area_Chair_h1JQ · 2025-12-19

**Summary:**

The paper proposes GHS-TDA, a novel framework that integrates global hypothesis graphs with topological data analysis to enhance LLM reasoning. Reviewers commended the method's originality and its strong empirical performance across multiple benchmarks compared to established baselines like ToT and AoT. During the rebuttal, the authors effectively addressed concerns regarding computational efficiency, demonstrating a superior accuracy-cost trade-off compared to structured alternatives. Therefore, I recommend acceptance.

That said, the authors are encouraged to add more theoretical discussion of why their method works, to enhance the paper's comprehensiveness and theoretical depth.

**Reviewer Concerns:**

Addressed:
1. Reviewer rfA4 noted the method was tested on only three models. The authors adds more experiments.
2. Reviewers were concerned about complexity and requested details on token usage and latency. The authors provided tables comparing LLM calls, token usage, and wall-clock time, showing this is not a serious problem.
3. Reviewers questioned how the distance weights and merge thresholds were chosen. The authors provided ablation studies showing that semantic features are the most important, while structural and uncertainty terms provide auxiliary benefits.

Outstanding:
1. Lack of theoretical ground of TDA. Reviewer sw5k argued that linking TDA features to "reasoning reliability" lacks a clear mathematical proof and is mostly empirical.
2. Reviewer sw5k questioned if the complex pipeline is truly necessary compared to simple methods like CoT-SC.

**Reviewer Scores:**

Reviewer Ucj2 may not change the score, because this reviewer has low confidence and only have some general questions.

Reviewer sw5k may not change the score, because this reviewer was the most critical and engaged extensively in the discussion. Despite the authors' detailed responses regarding efficiency and theoretical grounding, Reviewer sw5k explicitly stated in their final comment that "several core concerns... remain unaddressed".

Reviewer rfA4 may increase the score because this reviewer was already leaning positive but requested validation on newer/larger models and a detailed cost analysis. The authors responded comprehensively by adding experiments on new models and providing a breakdown of token/call costs.

Reviewer XVaz may not change the score, because although the authors answered a few questions, this reviewer hopes to know the theoretical analysis of the method, which was not addressed.

---

### Decision · Program_Chairs · 2026-01-26

Accept (Poster)